# Improving Convergence Guarantees of Random Subspace Second-order Algorithm for Nonconvex Optimization

Rei Higuchi [*1,2], Pierre-Louis Poirion [†2], and Akiko Takeda [‡1,2]

[1]Graduate School of Information Science and Technology, The University of Tokyo
[2]Center for Advanced Intelligence Project, RIKEN

## Abstract

In recent years, random subspace methods have been actively studied for large-dimensional nonconvex problems. Recent subspace methods have improved theoretical guarantees such as iteration complexity and local convergence rate while reducing computational costs by deriving descent directions in randomly selected low-dimensional subspaces. This paper proposes the Random Subspace Homogenized Trust Region (RSHTR) method with the best theoretical guarantees among random subspace algorithms for nonconvex optimization. RSHTR achieves an $\varepsilon$-approximate first-order stationary point in $O(\varepsilon^{-3/2})$ iterations, converging locally at a linear rate. Furthermore, under rank-deficient conditions, RSHTR satisfies $\varepsilon$-approximate second-order necessary conditions in $O(\varepsilon^{-3/2})$ iterations and exhibits a local quadratic convergence. Experiments on real-world datasets verify the benefits of RSHTR.

## 1 Introduction

We consider unconstrained nonconvex optimization problems as follows:

$$\min_{x \in \mathbb{R}^n} f(x), \tag{1.0.1}$$

where $f : \mathbb{R}^n \to \mathbb{R}$ is a possibly nonconvex $C^2$ function and bounded below. Nonconvex optimization problems are often encountered in real-world applications, such as training deep neural networks, and they are often high-dimensional in recent years. Therefore, there is a growing need for nonconvex optimization algorithms with low complexity relative to dimensionality.

Recently, subspace methods have been actively investigated for problems with large dimension $n$; they compute the search direction at each iteration on a low $s$-dimensional space (i.e., $s \ll n$), reducing the computational cost involved in gradient computation and Hessian matrices. Among these methods, the method using random projection, where the function $f$ is restricted at each iteration $k$ to a subspace $P_k^\top \mathbb{R}^s$ with the use of a random matrix $P_k \in \mathbb{R}^{s \times n}$, is referred to as random subspace method. When finding the search direction at the iterate $x_k$, $\min_{\tilde{d} \in \mathbb{R}^s} f(x_k + P_k^\top \tilde{d})$ can be considered instead of $\min_{d \in \mathbb{R}^n} f(x_k + d)$ in the random subspace method, and various solution methods can be derived depending on how a solution $\tilde{d}$ is found.

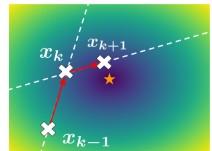

Figure 1: Illustration of our random subspace method on $\mathbb{R}^2$. Each iteration restricts the update to a 1-dim. randomly selected subspace.

---

[*]higuchi-rei714@g.ecc.u-tokyo.ac.jp

[†]pierre-louis.poirion@riken.jp

[‡]takeda@mist.i.u-tokyo.ac.jp, akiko.takeda@riken.jp

Table 1: Comparison of random subspace algorithms for nonconvex optimization. The SOSP column indicates whether the convergence point is a second-order stationary point or not. The "Feas." column indicates whether numerical experiments were given, implying the implementation is possible. The number in the "Local" column indicates the rate of convergence: 1 for linear, 1+ for superlinear, and 2 for quadratic. The † represents the condition that the Hessian at the local minimizer is rank deficient. The ∗ signifies that the objective function has low effective dimensionality. The ✓✓ indicates that in addition to the same assumption as the previous work in ✓, this property holds under another assumption.

| | Underlying algo. | Subprob. cost/iter | Global | Local | SOSP | Feas. |
|---|---|---|---|---|---|---|
| Roberts & Royer (2023) | Direct search | Multi. line-search | $O(\varepsilon^{-2})$ | | | ✓ |
| Dzahini & Wild (2024) | Zeroth order | Finite diff. grad. | $O(\varepsilon^{-2})$ | | | ✓ |
| Kozak et al. (2023) | Zeroth order | Finite diff. grad. | $O(\varepsilon^{-2})$ | 1 | | ✓ |
| Kozak et al. (2021) | Grad. descent | Gradient | $O(\varepsilon^{-2})$ | 1 | | ✓ |
| Cartis et al. (2020) | Gauss-Newton | Cond.quad.prog. (QP) | $O(\varepsilon^{-2})$ | | | ✓ |
| Shao (2022) | Cubic Newton | Cond. cubic.reg.QP | $O(\varepsilon^{-3/2})$ | | ✓ | |
| Zhao et al. (2024) | Cubic Newton | Cubic.reg.QP | $O(\varepsilon^{-3/2})$ | | ✓ | ✓ |
| Fuji et al. (2022) | Reg. Newton | Solve eq. | $O(\varepsilon^{-2})$ | $1+^{\dagger}$ | | ✓ |
| Ours | Trust Region | Min eigenvalue | $O(\varepsilon^{-3/2})$ | $2^{*}$ | ✓✓ | ✓ |

**Overview of existing random subspace methods:** We can use ideas from existing optimization algorithms for dealing with $\min_{\tilde{d} \in \mathbb{R}^s} f(x_k + P_k^\top \tilde{d})$, and various random subspace optimization methods have been proposed for convex optimization in e.g., Berahas et al. (2020); Gower et al. (2019); Grishchenko et al. (2021); Lacotte & Pilanci (2022); Pilanci & Wainwright (2014). Recently, some random subspace algorithms have started to be developed for nonconvex optimization and the main ones related to our research are summarized in Table 1. It may not be so challenging to construct a random subspace variant for each optimization method. What is difficult is to show that the sequence of iterates computed in subspaces converges with high probability to a stationary point of the original problem. The derivation of the faster local convergence rate (e.g., superlinear rate) is complicated when it comes to subspace methods. Indeed, Fuji et al. (2022) proved that even for strongly convex $f$ locally around a strict local minimizer, we cannot aim, with high probability, at local superlinear convergence using random subspace, even though the full dimensional regularized Newton method allows it. Based on the above and the discussion in Section 1.1 shown later, the following question naturally arises:

*To what extent can theoretical guarantees be improved by performing gradient-vector and Hessian-matrix operations in a low-dimensional subspace?*

**Our research idea:** Trust region methods are known to be fast and stable with excellent theoretical guarantees. Nevertheless, with conventional methods of solving subproblems, it is challenging to develop a subspace algorithm with convergence guarantees. In fact, prior to a recent series of subspace method studies, Erway & Gill (2010) proposed a trust region method based on projection calculations onto a low-dimensional subspace, but there was no discussion of convergence speed. To the best of our knowledge, random subspace trust region methods with complexity analysis have not been developed until now. Only recently, a new type of trust region method was developed by Zhang et al. (2022), and we realized that we could develop a subspace variant with convergence guarantees based on this method. However, deriving local convergence rates is still tricky. The trust region method in Zhang et al. (2022) treats the trust region subproblem as a basic minimum eigenvalue problem, which makes theoretical analysis possible. This approach contrasts with the intricate design of update rules in the previous trust region methods and makes the theoretical analysis of dimension reduction using the random subspace method more concise.

**Contribution:** We propose a new random subspace method, Random Subspace Homogenized Trust Region (RSHTR), which efficiently solves high-dimensional nonconvex optimization problems by identifying descent directions within randomly selected subspaces. RSHTR does not need to compute the restricted Hessian, $P\nabla^2 f(x)P^\top \in \mathbb{R}^{s \times s}$, making our algorithm more advantageous than other second-order methods numerically. It also has excellent theoretical properties as in Table 1: more concretely,

- convergence to an $\varepsilon$–approximate first-order stationary point with a global convergence rate of $O(\varepsilon^{-3/2})$, giving an analysis of the total computational complexity and confirming that the total computational complexity, as well as space complexity, are improved over the existing algorithm (Zhang et al., 2022) in full space,
- convergence to a second-order stationary point under the assumption that the Hessian at the point is rank-deficient or under the same assumption as Shao (2022),
- local quadratic convergence under the assumption on $f$ being strongly convex within its low effective subspace.

This rank-deficient assumption includes cases commonly observed in recent machine-learning optimization problems. Indeed, the structure of the Hessian in neural networks has been studied both theoretically and experimentally, revealing that it often possesses a low-rank structure (Wu et al., 2020; Sagun et al., 2017; 2016; Ghorbani et al., 2019). Thus, our theoretical guarantee is considered important from a practical perspective.

### 1.1 Existing random subspace algorithms for nonconvex optimization

As summarized in Table 1, various types of random subspace methods have been proposed in recent years to tackle high-dimensional machine learning applications characterized by over-parametrization. Several papers (Hanzely et al., 2020; Shao, 2022) investigate subspace cubic regularization algorithm. It can be noticed that Shao (2022) achieves the current best global convergence rate for a random subspace method of $O(\varepsilon^{-3/2})$ for a nonconvex function. Although it guarantees the convergence to a second-order stationary point under some strong assumptions on the condition number and the rank of the Hessian around that point, it does not discuss local convergence rates. We also noticed a concurrent work (Zhao et al., 2024), which was uploaded after finishing our project, ensuring the convergence rate of $O(\varepsilon^{-3/2})$ to a second-order stationary point. However, their algorithm requires a subspace size $s = \Omega(n)$ in general for the guarantee, which is larger than ours $s = \Omega(\log n)$, and leads to larger computation cost per iteration (see Theorem 3.2).

In contrast, the study by Fuji et al. (2022) proves that a subspace variant of the regularized Newton method achieves linear local convergence in general and superlinear local convergence under the rank-deficiency assumption for the Hessian. However, the global convergence rate is limited to $O(\varepsilon^{-2})$, and they only guarantee the convergence to a first-order stationary point. Other random subspace algorithms for nonconvex optimization include Roberts & Royer (2023), where the algorithm converges to first-order stationary points with $O(\varepsilon^{-2})$ and the local convergence rates are not studied. A series of studies (Cartis et al., 2022; 2023; Cartis & Otemissov, 2021) assume the use of a global optimization method to solve subproblems, and do not discuss local convergence rates.

**Notations.** We define $I$ as the identity matrix. For related quantities $\alpha$ and $\beta$, we write $\alpha = O(\beta)$ if there exists a constant $c > 0$ such that $\alpha \leq c\beta$ for all $\beta$ sufficiently small. We also write $\alpha = o(\beta)$ if $\lim_{\beta \to 0} \frac{\alpha}{\beta} = 0$ holds, and $\alpha = \Omega(\beta)$ if $\beta = O(\alpha)$. For a positive semi-definite matrix $S$, we denote by $\sqrt{S}$ its squared root, i.e., $\sqrt{S}\sqrt{S} = S$.

## 2 Proposed method

### 2.1 Existing algorithm: HSODM

Before proposing our method, we briefly describe a new type of trust region method, a homogeneous second-order descent method (HSODM), for the nonconvex optimization problem (1.0.1); see Appendix A for the details. One of the exciting points of HSODM (Zhang et al., 2022) is that it uses eigenvalue computations to solve trust region subproblems (TRSs). The TRS is a subproblem constructed and solved in each iteration of the trust region method. Various solution methods have been proposed for TRS so far, and due to the improvement in eigenvalue computation, solving the TRS with eigenvalue computation has recently attracted much attention (see, e.g., Adachi et al. (2017); Lieder (2020)). HSODM is a method that incorporates the idea of homogenization into TRS and solves it by eigenvalue computation in the trust region method.

## 2.2 Random Subspace Homogenized Trust Region: RSHTR

Now we propose Random Subspace Homogenized Trust Region (RSHTR), which is an algorithm that reduces the dimension of the subproblems solved in HSODM with random projection and solves it by eigenvalue computations.

For the sake of detail, we will first define by $\tilde{f}_k$ the function $f$ restricted to a low $s$-dimensional subspace $P_k^\top \mathbb{R}^s$ (i.e., $s \ll n$) containing the current iterate $x_k$, that is, $\forall u \in \mathbb{R}^s$, $\tilde{f}_k(u) := f(x_k + P_k^\top u)$, using a random matrix $P_k \in \mathbb{R}^{s \times n}$, where each element follows an independent normal distribution $\mathcal{N}(0, 1/s)$. Random Gaussian matrices $P_k$ are sampled independently at each iteration $k$. Using the notation $g_k := \nabla f(x_k)$ and $H_k := \nabla^2 f(x_k)$, we can write the gradient and Hessian matrix of $\tilde{f}_k$ as $\tilde{g}_k := P_k g_k, \tilde{H}_k := P_k H_k P_k^\top$. The tilde symbol denotes the subspace counterpart of the corresponding variable. Using the notation, the subproblem we need to solve at each iteration can be written as:

$$\min_{\|[\tilde{v};t]\| \le 1} \begin{bmatrix} \tilde{v} \\ t \end{bmatrix}^\top \begin{bmatrix} \tilde{H}_k & \tilde{g}_k \\ \tilde{g}_k^\top & -\delta \end{bmatrix} \begin{bmatrix} \tilde{v} \\ t \end{bmatrix}, \tag{2.2.1}$$

where $\delta \ge 0$ is a parameter appropriately selected to meet the desired accuracy. The relation between $\delta$ and the accuracy will be shown in Section 3. This subproblem is a lower dimensional version of the subproblem solved in Zhang et al. (2022). We explain in Appendix A the motivation of the above subproblem. Similar to the subproblem solved in Zhang et al. (2022), (2.2.1) can also be regarded as a problem of finding the leftmost eigenvector of a matrix. Hence, (2.2.1) is solved using eigenvalue solvers, such as Lanczos tridiagonalization algorithm (Golub & Van Loan, 2013) and the randomized Lanczos algorithm (Kuczyński & Woźniakowski, 1992). The random projection decreases the subproblem dimension from $n+1$ to $s+1$. Therefore, unlike HSODM, RSHTR allows for other eigensolvers beyond the Lanczos method.

Using the solution of (2.2.1) denoted by $[\tilde{v}_k, t_k]$, we can approximate the solution of the full-space subproblem by $[P_k^\top \tilde{v}_k, t_k]$. We then define the descent direction $d_k$ and the iterates update as[1]

$$d_k := \left\{ \begin{array}{ll} P_k^\top \tilde{v}_k / t_k, & \text{if } t_k \ne 0 \\ P_k^\top \tilde{v}_k, & \text{otherwise} \end{array} \right. \quad \& \quad x_{k+1} = \left\{ \begin{array}{ll} x_k + \eta_k d_k, & \text{if } \|d_k\| > \Delta \\ x_k + d_k, & \text{if } \|d_k\| \le \Delta \end{array} \right. . \tag{2.2.2}$$

In RSHTR, the step size selection is fixed to $\eta_k = \Delta/\|d_k\|$, and we do not address line search strategy. However, it should be noted that it is also possible to give theoretical guarantees when we adopt a usual line search strategy in RSHTR to compute $\eta_k$. The complete algorithm is given in Algorithm 1.

As shown in Section 3, the output $\hat{x}$ of our algorithm is an $\varepsilon$–approximate first-order stationary point ($\varepsilon$–FOSP), i.e., it satisfies $\|\nabla f(\hat{x})\| \le O(\varepsilon)$. If some assumption is satisfied, it will be an $\varepsilon$–approximate second-order stationary point ($\varepsilon$–SOSP), i.e., it satisfies $\lambda_{\min}(\nabla^2 f(\hat{x})) \ge \Omega(-\sqrt{\varepsilon})$. More precisely,

- if the condition $\|d_k\| \le \Delta$ is satisfied, the algorithm can either terminate and output $x_{k+1}$, thereby obtaining an $\varepsilon$–FOSP (or $\varepsilon$–SOSP if stronger Assumption 2 holds),
- or reset $\delta = 0$, fix the update rule to $x_{k+1} = x_k + d_k$ and continue, thereby achieving local linear convergence (or quadratic convergence if stronger Assumption 4 holds).

## 2.3 Total computational complexity and space complexity

We also discuss the computational cost per iteration in Algorithm 1. The main cost involving $P_k$ and $P_k^\top$ is dominated by the computation of $P_k H_k P_k^\top v$ for a vector $v$ in Line 6 of Algorithm 1. Indeed, we notice that it is not needed to compute the whole matrix $P_k H_k P_k^\top$

---

[1] It might seem to be more natural to describe $|t_k| > \nu$ instead of $t_k \ne 0$ as in HSODM (see Algorithm 2) for pure random subspace variant of HSODM. However, in the fixed-radius strategy, on which we focus (see (2.2.2)), even if we obtain theoretical guarantees for any $\nu \in (0, 1/2)$, the results do not depend on $\nu$. Therefore, we discuss the case with $\nu \to +0$ for simplicity and clarity. Theoretical guarantees regarding a general $\nu$ are provided in Appendix C.

---

**Algorithm 1** RSHTR: Random Subspace Homogenized Trust Region Method

---

1: **function** $\mathrm{RSHTR}(s, n, \delta, \Delta, \mathrm{max\_iter})$
2:     `global_mode = True`
3:     **for** $k = 1, \ldots, \mathrm{max\_iter}$ **do**
4:         $P_k \leftarrow s \times n$ random Gaussian matrix with each element being from $\mathcal{N}(0, 1/s)$
5:         $\tilde{g}_k \leftarrow P_k g_k$
6:         $(t_k, \tilde{v}_k) \leftarrow$ optimal solution of (2.2.1) by eigenvalue computation
7:         $d_k \leftarrow \begin{cases} P_k^\top \tilde{v}_k / t_k, & \text{if } t_k \neq 0 \\ P_k^\top \tilde{v}_k, & \text{otherwise} \end{cases}$
8:         **if** `global_mode` and $\|d_k\| > \Delta$ **then**
9:             $\eta_k \leftarrow \Delta/\|d_k\|$                     ▷ or get from backtracking line search
10:             $x_{k+1} \leftarrow x_k + \eta_k d_k$
11:         **else**
12:             **terminate** ▷ or continue with $(\delta, \texttt{global\_mode}) \leftarrow (0, \texttt{False})$ for local conv.
13:             $x_{k+1} \leftarrow x_k + d_k$

---

in order to solve the subproblem (2.2.1) by the Lanczos algorithm. This can be done by computing the Hessian-vector product $\nabla^2 \tilde{f}_k(0) v$ where we recall that $\tilde{f}_k(u) = f(x_k + P_k^\top u)$. Assuming the use of the Hessian-vector product operation, we discuss below the computation cost of Line 6, as well as the total computational complexity of Algorithm 1.

Assuming that the full-space Hessian-vector product can be computed in $O(n)$ using back-propagation (see (Pearlmutter, 1994)), the total complexity of computing $\nabla^2 \tilde{f}_k(0) v$ becomes $O(sn)$. Furthermore, Lanczos tridiagonalization algorithm solves, using a random initialization, (2.2.1) exactly (Golub & Van Loan, 2013, Theorem 10.1.1). The computational cost boils down to computing $s + 1$ matrix vector products. Hence, by using Hessian-vector product, the computational complexity to solve (2.2.1) exactly is $(s+1) \cdot O(sn) = O\left(s^2 n\right)$. Furthermore, the iteration complexity, under the gradient and Hessian Lipschitz assumptions (Assumption 1) and the low effective setting (Assumption 4), is $O((n/s)^{3/4} \varepsilon^{-3/2})$ as detailed in Section 3.1. Consequently, the total computational complexity of RSHTR in this setting is $O(\varepsilon^{-3/2} s^{5/4} n^{7/4})$. On the other hand, the total computational complexity of the full-space algorithm HSODM (Zhang et al., 2022) is $O(\varepsilon^{-3/2} n^2)$ because the complexity of solving exactly the subproblem, using Hessian vector products, becomes $O(n^2)$. Therefore, when $s = o(n)$ (for example $s = O(\log(n))$), the total computational complexity of the proposed method is smaller than that of HSODM. Notice that in any case (for any value of $s < n$) the actual execution time of the proposed method outperforms HSODM. This is because HSODM's space complexity[2] explodes to $O(n^2)$ due to the Hessian, whereas the proposed method's space complexity is limited to $O(sn)$ taking into account that the $s \times n$ random matrix is larger than the restricted Hessian.

## 3 THEORETICAL ANALYSIS

We analyze the global and local convergence properties of RSHTR. First, we show that RSHTR converges, with high probability, to an $\varepsilon$–FOSP in at most $O(\varepsilon^{-3/2})$ iterations. Secondly, we prove that, under some conditions, the algorithm actually converges to an $\varepsilon$–SOSP. Lastly, we investigate local convergence: proving local linear convergence and, under a rank deficiency condition, local quadratic convergence. Note that the analysis in this section is inspired by Zhang et al. (2022).

**Assumption 1.** $f$ has $L$-Lipschitz continuous gradient and $M$-Lipschitz continuous Hessian, that is, for all $x, y \in \mathbb{R}^n$,

$$\|\nabla f(x) - \nabla f(y)\| \leq L\|x - y\|, \quad \left\|\nabla^2 f(x) - \nabla^2 f(y)\right\| \leq M\|x - y\|.$$

---

[2]Here, we assume the Hessian matrix is stored because rigorously evaluating the space complexity of the Hessian-vector product is challenging.

To prove the global convergence property, we first state that when $\|d_k\| > \Delta$ holds, the objective function value decreases sufficiently at each iteration.

**Lemma 3.1.** *Suppose that Assumption 1 holds. If $\|d_k\| > \Delta$, then for all $\delta > 0$*

$$f(x_{k+1}) - f(x_k) \leq -\frac{1}{2(\sqrt{n/s}+\mathcal{C})^2}\Delta^2\delta + \frac{M}{6}\Delta^3$$

*with probability at least $1 - 2\exp(-s)$. Here $\mathcal{C}^3$ is an absolute constant.*

The following lemma shows that if the descent direction once get small enough, the norm of the gradient in the subsequent iteration is bounded using $\delta$ and $\Delta$.

**Lemma 3.2.** *Suppose that Assumption 1 holds. If $\|d_k\| \leq \Delta \leq \frac{1}{2\sqrt{2}}$, then*

$$\Pr\left[\|g_k\| \leq 4\Delta\left(L\sqrt{n/s+\mathcal{C}} + \frac{\Delta\delta}{\sqrt{n/s-\mathcal{C}}}\right)\right] \geq 1 - 2\exp\left(-\frac{C}{4}s\right) - 2\exp(-s).$$

Notice that we can assume that $\Delta \leq \frac{1}{2\sqrt{2}}$ holds w.l.o.g. as we want to take $\Delta$ as small as possible to ensure a better convergence rate. Next, we show a lower bound on the minimum eigenvalue of the Hessian.

**Lemma 3.3.** *Suppose that Assumption 1 holds. If $\|d_k\| \leq \Delta \leq \frac{1}{2\sqrt{2}}$, then we have*

$$\Pr\left[\tilde{H}_k \succeq -\left(8\Delta^2\left(\left(\sqrt{\frac{n}{s}}+\mathcal{C}\right)^2 L+\delta\right)+\delta\right)I\right] \geq 1 - 2\exp\left(-\frac{C}{4}s\right) - 2\exp(-s).$$

Although Lemma 3.3 does not directly provide a lower bound on the eigenvalues of $H_k$, it is proportional to the bound in Lemma 3.3 under certain conditions. We later discuss this in Section 3.2.

### 3.1 GLOBAL CONVERGENCE TO AN $\varepsilon$–FOSP

We now prove that our algorithm converges to an $\varepsilon$–FOSP under a general assumption.

**Theorem 3.1** (Global convergence to an $\varepsilon$–FOSP)**.** *Suppose that Assumption 1 holds. Let*

$$0 < \varepsilon \leq \frac{M^2}{8}, \quad \delta = \left(\sqrt{\frac{n}{s}}+\mathcal{C}\right)^2\sqrt{\varepsilon} \quad and \quad \Delta = \frac{\sqrt{\varepsilon}}{M}. \qquad (3.1.1)$$

*If there exists a positive constant $\tau$ such that $s = O(n\varepsilon^{1/\tau})$, then RSHTR outputs an $\varepsilon$–FOSP in at most $O\left(\varepsilon^{-3/2}\right)$ iterations with probability at least*

$$1 - 4\exp\left(-Cs/4\right) - (2U_\varepsilon + 2)\exp\left(-s\right), \qquad (3.1.2)$$

*where $\mathcal{C}$ and $C$ are absolute constants and $U_\varepsilon := \lfloor 3M^2\left(f(x_0) - \inf_{x\in\mathbb{R}^n} f(x)\right)\varepsilon^{-3/2}\rfloor + 1$.*

Theorem 3.1 leads to the following corollary on the probability bound, which is confirmed by rewriting (3.1.2) as $1 - 4n^{-Cc/4} - (2U_\varepsilon + 2)n^{-c}$ with $s = c\log n$ for some $c > 0$. Notice that $s = O(n\varepsilon^{1/\tau})$ holds for a small value of $\tau$ if $n$ is large and $s$ is not too small.

**Corollary 3.1.** *If $s = \Omega(\log n)$, then the probability bound (3.1.2) is in the order of $1-o(1)$.*

For comparison with the full-space algorithm HSODM (Zhang et al., 2022), when considering the dependency on $n$ and $s$, the norm of the gradient at the output point is $O((n/s)^{(2+\tau)/2}\varepsilon)$. This result is obtained by substituting the given parameters into Lemma 3.2. This is $O((n/s)^{(2+\tau)/2})$ times worse than the HSODM. Since we want to obtain an $\varepsilon$–FOSP instead of an $((n/s)^{(2+\tau)/2}\varepsilon)$–FOSP, we need to scale down $\varepsilon$ by $O((n/s)^{-(2+\tau)/2})$ leading the number of iterations increased by a factor $(n/s)^{\frac{3}{2}+\frac{3\tau}{4}}$. We have added some detailed explanations in Appendix D.3 just after the proof of Theorem 3.1. This is the price to pay for utilizing random subspace.

We should emphasize that under the assumption of low-effectiveness (Assumption 4) introduced in Section 3.4, the exponent of the $n/s$ factor can be improved to $3/4$. Therefore, the total complexity of RSHTR becomes smaller than that of HSODM as discussed in Section 2.3. See Appendix D.4 for more details.

As mentioned in the footnote 1, these results (essentially Lemma 3.1) can also be demonstrated for any $\nu \in (0, 1/2)$, i.e., the similar statement also holds for the pure random subspace variant of HSODM shown in Algorithm 3. Refer to Appendix C for the proof.

---

[3]See Wainwright (2019) for more details.

## 3.2 Global convergence to an $\varepsilon$–SOSP

Section 3.1 demonstrated that RSHTR globally converges to an $\varepsilon$–FOSP. In this section, we show that the output $x^*$ of RSHTR is also an $\varepsilon$–SOSP under one of two possible assumptions about the Hessian at the point, one of which is presented in Shao (2022). For clarity and brevity, we move the result under the assumption from Shao (2022) in Appendix D.5 while focusing on the more practical one in this section.

**Assumption 2.** Let $r = \mathrm{rank}(\nabla^2 f(x^*))$ for the output $x^*$ of RSHTR. We assume $r \leq s$.

Notice that this includes many cases commonly observed in recent machine-learning optimization problems. Indeed, the structure of the Hessian in neural networks has been studied both theoretically and experimentally, revealing that it often possesses a low-rank structure (Wu et al., 2020; Sagun et al., 2017; 2016; Ghorbani et al., 2019). We also introduce some problems satisfying a stronger condition than Assumption 2 at the beginning of Section 3.4. Thus, our theoretical guarantee is considered important from a practical perspective. We guarantee in Theorem 3.2 that the output $x^*$ of RSHTR is a $\varepsilon$–SOSP when the Hessian at $x^*$, denoted by $H^*$, is rank deficient. Before proceeding to the theorem, we show in Lemma 3.4 that the lower bound of the minimum eigenvalue of $H^*$ is proportional to the lower bound of the minimum eigenvalue of $P^* H^* P^{*\top}$ or non-negative with high probability. Here $P^*$ denotes the matrix $P_k$ used at the last iteration of the algorithm.

**Lemma 3.4.** *Let $\bar{C}$ and $\bar{c}$ be absolute constants[4]. If Assumption 2 holds, then for all $\zeta > 0$, the following inequality holds with probability at least $1 - (\bar{C}\zeta)^{s-r+1} - e^{-\bar{c}s}$.*

$$\lambda_{\min}(H^*) \geq \zeta^{-2} \left(1 - \sqrt{(r-1)/s}\right)^{-2} \min\left\{\lambda_{\min}\left(P^* H^* P^{*\top}\right), 0\right\}.$$

This lemma implies that under Assumption 2, it is sufficient to guarantee that the output of RSHTR is $\varepsilon$–SOSP in full space if it is $\varepsilon$–SOSP in a subspace. Since the minimum eigenvalue of $H^*$ is bounded as shown in Lemma 3.3, the following theorem holds.

**Theorem 3.2** (Global convergence to an $\varepsilon$–SOSP under rank deficiency). *Suppose that Assumptions 1 and 2 hold. Set $\varepsilon, \delta$ and $\Delta$ as in (3.1.1) of Theorem 3.1. If there exists a positive constant $\tau$ such that $s = O(n\varepsilon^{1/\tau})$, then RSHTR outputs an $\varepsilon$–SOSP in at most $O(\varepsilon^{-3/2})$ iterations with probability at least*

$$1 - 6\exp\left(-Cs/4\right) - (2U_\varepsilon + 4)\exp\left(-s\right) - \exp\left(-s + r - 1\right) - \exp(-\bar{c}s),$$

*where $\bar{c}$ and $C$ are absolute constants and $U_\varepsilon := \lfloor 3M^2 \left(f(x_0) - \inf_{x \in \mathbb{R}^n} f\right) \varepsilon^{-3/2}\rfloor + 1$.*

For comparison with the full-space algorithm HSODM (Zhang et al., 2022), when considering the dependency on $n$ and $s$, the minimum eigenvalue of the Hessian at the output point is $\Omega(-(n/s)\sqrt{\varepsilon})$. This is $n/s$ times worse than the full-space method. In other words, the number of iterations required to achieve the same accuracy as the full-space algorithm is $(n/s)^3$ times larger. Unlike convergence to an $\varepsilon$–FOSP, convergence to an $\varepsilon$–SOSP requires an additional assumption, Assumption 2. The convergence guarantee to $\varepsilon$–SOSP is not easy because the algorithm is run until the termination condition is satisfied, considering only subspaces, while convergence to $\varepsilon$–SOSP can be shown without any extra assumption thanks to Johnson-Lindenstrauss lemma, Lemma B.1. Recall that other random subspace algorithms (Zhao et al., 2024; Shao, 2022) also showed convergence to an $\varepsilon$–SOSP, but Zhao et al. (2024) required the dimension of the subspace $s$ to be $\Omega(n)$ in general, and Shao (2022) required stronger assumptions as explained at the beginning of this subsection.

## 3.3 Local linear convergence

In this section, we discuss the local convergence of RSHTR to a strict local minimizer $\bar{x}$. Here, we consider the case when we continue to run Algorithm 1 after $\|d_k\| \leq \Delta$ is satisfied, by setting $\delta = 0$ and `global_mode = False`. We consider a standard assumption for local convergence analysis.

**Assumption 3.** Assume that RSHTR converges to a strict local minimizer $\bar{x}$ such that $\nabla f(\bar{x}) = 0, \nabla^2 f(\bar{x}) \succ 0$.

---

[4]We refer the reader to Rudelson & Vershynin (2009) for estimations on these constants.

We denote the Hessian at $\bar{x}$ by $\bar{H}$ and introduce the norm $\|x\|_{\bar{H}} = \sqrt{x^\top \bar{H} x}$. This assumption implies that for any $\delta > 0$, there exists $k_0$ such that $\frac{o(\|x_k - \bar{x}\|)}{\|x_k - \bar{x}\|} \leq \delta$ for $\forall k \geq k_0$.

**Theorem 3.3** (Local linear convergence). *Suppose Assumptions 1 and 3 hold. Then, for $k$ large enough, i.e., there exists $k_0$ such that for all $k \geq k_0$,*

$$\Pr\left[\|x_{k+1} - \bar{x}\|_{\bar{H}} \leq \sqrt{1 - \frac{1}{4\kappa(\bar{H})(\sqrt{n/s} + C)^2}}\, \|x_k - \bar{x}\|_{\bar{H}}\right] \geq 1 - 6\exp(-s),$$

*where $\kappa(\bar{H}) := \lambda_{\max}(\bar{H})/\lambda_{\min}(\bar{H})$.*

Trust region method closely resembles Newton's method in a sufficiently small neighborhood of a local minimizer. Consequently, we expect that the impossibility of achieving local superlinear convergence for RSHTR in general can be shown, similar to Fuji et al. (2022). Indeed, in order to prove local superlinear convergence, Zhang et al. (2022) decompose the direction $d_k = d_k^N + r_k$, where $d_k^N$ corresponds to the direction in a Newton step. In the subspace setting, this strategy would fail because the $d_k^N$ part of the descent direction $d_k$ would hinder superlinear convergence, as proved in Fuji et al. (2022).

### 3.4 LOCAL CONVERGENCE FOR STRONGLY CONVEX $f$ IN ITS EFFECTIVE SUBSPACE

We now consider the possibility of our algorithm achieving local quadratic convergence by making stronger assumptions than Assumption 3 on the function $f$. Concretely, we focus on so-called "functions with low dimensionality"[5] (Wang et al., 2016), which satisfy the following condition:

$$\exists \Pi \in \mathbb{R}^{n \times n}, \ \text{rank}(\Pi) \leq n - 1, \ \text{ s.t. } \ \forall x \in \mathbb{R}^n, \ f(x) = f(\Pi x), \tag{3.4.1}$$

where $\Pi$ is an orthogonal projection matrix. These are the functions that only vary over a low-dimensional subspace (which is not necessarily be aligned with standard axes), and remain constant along its orthogonal complement. Such functions are frequently encountered in many applications. For instance, the loss functions of neural networks often have low rank Hessians Gur-Ari et al. (2018); Sagun et al. (2017); Papyan (2018). This phenomenon is also prevalent in other areas such as hyper-parameter optimization for neural networks (Bergstra & Bengio, 2012), heuristic algorithms for combinatorial optimization problems (Hutter et al., 2014), complex engineering and physical simulation problems as in climate modeling (Knight et al., 2007), and policy search (Fröhlich et al., 2019).

Now we show a stronger assumption than Assumption 3 on the function $f$.

**Assumption 4.** $f$ has $s$-low effective dimensionality as defined in (3.4.1) with an additional restriction $\text{rank}(\Pi) \leq s$. Furthermore, $f$ is $\rho$–strongly convex within its effective subspace.

The assumption indicates that for $R \in \mathbb{R}^{\text{rank}(\Pi) \times n}$ being a matrix whose columns form an orthonormal basis for $\text{Im}(\Pi)$, the function $l(y) := f(R^\top y)$ is $\rho$–strongly convex. To measure the distance within the effective subspace, we use the semi-norm $\|x\|_\Pi = \sqrt{\|x^\top \Pi x\|} = \|Rx\|$.

Now we show the local quadratic convergence property of RSHTR with parameters $\delta$ and `global_mode` being reset to 0 and `False`, respectively, similarly to the local convergence discussion in Section 3.3. This is the first theoretical result of random subspace methods having quadratic convergence properties for some classes of functions.

**Theorem 3.4** (Local quadratic convergence under $\rho$–strong convexity in effective subspace). *Suppose Assumptions 1 and 4 hold. Then, for $k$ large enough, i.e., there exists $k_0$ such that for all $k \geq k_0$, the following inequalities hold:*

$$\Pr\left[\|x_{k+1} - \bar{x}\|_\Pi \leq 4M_l \|R\|\rho^{-1}\sigma_{\min}(R^\top)^{-1}\|x_k - \bar{x}\|_\Pi^2\right] \geq 1 - 3e^{-r} - e^{-\bar{c}s},$$

$$\Pr\left[f(x_{k+1}) - f(\bar{x}) \leq 8L_l M_l^2 \|R\|^2 \rho^{-4}\sigma_{\min}(R^\top)^{-2}\left(f(x_k) - f(\bar{x})\right)^2\right] \geq 1 - 3e^{-r} - e^{-\bar{c}s},$$

*where $r = \text{rank}(\Pi)$, $\bar{c}$ is a universal constant, $L_l$ and $M_l$ are the Lipschitz constants of $\nabla l$ and $\nabla^2 l$ respectively, and $\bar{x}$ is the strict local minimizer of $f$.*

---

[5]They are also called objectives with "active subspaces" (Constantine et al., 2014), or "multi-ridge" (Fornasier et al., 2012).

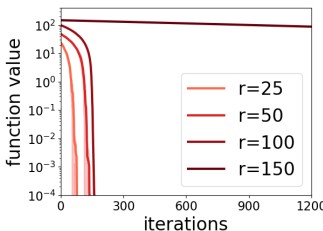
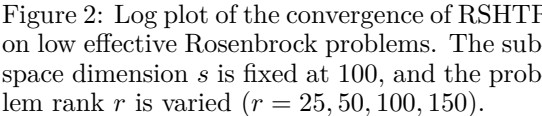
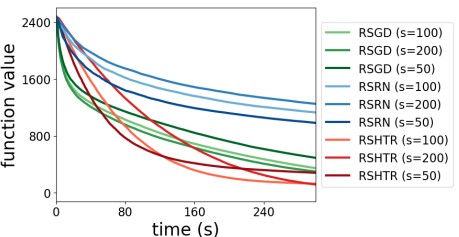

Figure 2: Log plot of the convergence of RSHTR on low effective Rosenbrock problems. The subspace dimension $s$ is fixed at 100, and the problem rank $r$ is varied ($r = 25, 50, 100, 150$).

Figure 3: The impact of the choice of subspace dimension $s$ ($= 50, 100, 200$) on convergence in random subspace algorithms (RSGD, RSRN, RSHTR) for MF.

## 4 NUMERICAL EXPERIMENTS

We compare the performance of our algorithm and existing methods: HSODM (Zhang et al., 2022), RSRN (Fuji et al., 2022), Gradient Descent (GD), and Random Subspace Gradient Descent (RSGD) (Kozak et al., 2021). We used backtracking line search in all algorithms to determine the step size. Unless otherwise noted, the subspace dimension $s$ is set to 100 for all the algorithms utilizing random subspace techniques. In HSODM and our method, we solve subproblems using the Lanczos method. The numerical experiments were conducted in the environment: CPU: Intel(R) Xeon(R) CPU E5-2697 v2 @ 2.70GHz, GPU: NVIDIA RTX A5000, RAM: 32 GB. The details of datasets we used are provided in Appendix F.

**Low Effective Rosenbrock function (LER):**  To illustrate the theoretical properties proved in this paper, we conducted numerical experiments on a Low Effective Rosenbrock (LER) function, chosen for its property of satisfying Assumptions 1 and 2. This function is defined as $\min_{x \in \mathbb{R}^n} R(A^\top A x)$, where $R(x) = \sum_{i=1}^{n-1} 100(x_{i+1} - x_i^2)^2 + (x_i - 1)^2$ and $A \in \mathbb{R}^{r \times n}$ with $r < n$. We set $n = 10000$ to represent a high-dimensional setting.

Figure 2 shows experiments varying $r$ with fixed $s$. When the rank deficiency $r \le s$ holds, we observe the predicted quadratic convergence. However, when $r > s$, convergence slowed significantly. In Figure 4a, we set $r = 50$ ($\le s = 100$). The results show that the full-space algorithm did not complete even a single step (and is thus omitted from the figure), while our algorithm outperformed the other algorithms, which supports the discussion in Section 2.3. Moreover, our method surpasses the other random subspace methods, consistently achieving the fastest global convergence rate.

**Matrix factorization (MF):**  We evaluate the real-world performance for MF using MovieLens 100k (Harper & Konstan, 2015): $\min_{U \in \mathbb{R}^{n_u \times k}, V \in \mathbb{R}^{k \times n_v}} \|UV - R\|_F^2 / (n_u n_v)$, where $R \in \mathbb{R}^{n_u \times n_v}$ and $\|\cdot\|_F$ denotes the Frobenius norm.

We first examine the effect of changing $s$ on each algorithm. Figure 3 shows that despite varying $s$, the relative performance among these methods remained consistent, and no particular method benefited from specific subspace dimensions. This finding justifies our choice of a fixed subspace dimension ($s = 100$) for all subspace methods in all the other experiments. Next, the performance of our proposed method is compared against other algorithms, as shown in Figure 4b. Although RSGD initially exhibits the fastest decrease, likely due to its advantage of not using the Hessian, our method soon surpasses RSGD.

**Classification:**  We test classification tasks using cross-entropy loss:

$$\min_{w \in \mathbb{R}^n} \quad -\frac{1}{N} \sum_{i=1}^{N} \sum_{j=1}^{K} \mathbb{I}[y_i = j] \log \left( \exp(\phi_w^j(x_i)) \Big/ \sum_{k=1}^{K} \exp(\phi_w^k(x_i)) \right),$$

where $N$ is the number of data and its label, $K$ is the number of classes in the classification, $(x_i, y_i)$ is the $i$-th data, and $\phi_w(x_i) = (\phi_w^1(x_i), \ldots, \phi_w^K(x_i))$ is the model's predicted logit. We explore the following specific instances.

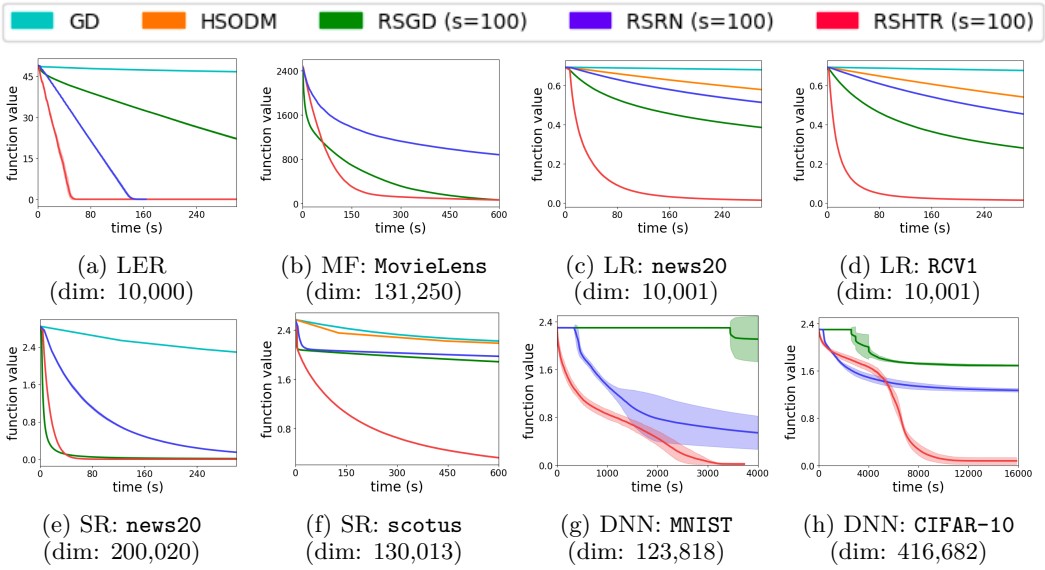

Figure 4: Comparison of our method to existing methods regarding the function value v.s. computation time. Each plot shows the average $\pm$ the standard deviation for five runs. Algorithms that did not complete a single iteration within the time limit are omitted.

- Logistic Regression (LR): $K = 2$ and $\phi_w^1(x_i) = w^T[x_i; 1]$ (with adaptation for the second class). News20 (Lang, 1995) and RCV1 datasets (Lewis et al., 2004) were used.
- Softmax Regression (SR): $K > 2$ and $\phi_w^j(x_i) = w_j^T[x_i; 1]$. News20 (Lang, 1995) and SCOTUS (Chalkidis et al., 2021) datasets were used.
- Deep Neural Networks (DNN): $\phi_w(x_i)$ represents the output of a 16-layer fully connected neural network. MNIST (Deng, 2012) and CIFAR-10 (Krizhevsky, 2009) were used.

In all the experiments (Figures 4c, 4d, 4e, 4f, 4g, 4h) and additional ones in Appendix G, our algorithm outperforms existing methods. A notable feature of our method is its rapid escape from flat regions. This can be attributed to the algorithm's second-order nature, the homogenization of the subproblem and the utilization of random subspace techniques. In addition, as discussed in the introduction, the rank-deficient assumptions (Assumptions 2 and 4) reflect scenarios commonly encountered in modern machine learning optimization problems. Consequently, our method is well-suited to applying Theorems 3.2 and 3.4.

## 5 FUTURE WORK

We proposed a new random subspace trust region method and confirmed its usefulness theoretically and practically. We believe that our proposed method can be made faster by incorporating various techniques if theoretical guarantees such as convergence rate are not required. For example, the parameter $\delta$ is fixed, once and for all, at the beginning of the algorithm. In some future work, it would be interesting to develop an adaptive version of this algorithm where the parameter $\delta > 0$ adapts to the current iterate.

Recently, authors of HSODM (Zhang et al., 2022) have been vigorously using the idea of HSODM to develop various variants (He et al., 2023) of HSODM, application to stochastic optimization (Tan et al., 2023), and generalization of the trust region method (Jiang et al., 2023). We want to investigate whether random subspace methods can be developed similarly.

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

## A    Existing work: HSODM

Homogeneous Second-Order Descent Method (HSODM) proposed by Zhang et al. (2022) is a type of trust region method, which globally converges to an $\varepsilon$–SOSP at a rate of $O(\varepsilon^{-3/2})$ and locally converges at a quadratic rate. The algorithm determines the descent direction based on the solution of the eigenvalue problem obtained by homogenizing the trust region subproblem. The algorithm procedure is described below.

At each iteration, HSODM minimizes the homogenized quadratic model. In other words, it solves the following subproblem:

$$\min_{\|[v;t]\|\leq 1} \begin{bmatrix} v \\ t \end{bmatrix}^\top \begin{bmatrix} H_k & g_k \\ g_k^\top & -\delta \end{bmatrix} \begin{bmatrix} v \\ t \end{bmatrix}, \tag{A.0.1}$$

where $\delta \geq 0$ is a parameter appropriately determined according to the required accuracy. The motivation behind is to force the Hessian matrix $H_k$ to have negative curvature. To do that (see Ye & Zhang (2003)), we homogenize the second order Taylor expansion, $m_k(d)$, of $f(x_k + d)$:

$$m_k(d) = g_k^\top d + \frac{1}{2} d^\top H_k d.$$

By rewriting $d = \frac{v}{t}$, we have

$$t^2 \left( m_k(d) - \frac{1}{2}\delta \right) = t^2 \left( g_k^\top(v/t) + \frac{1}{2}(v/t)^\top H_k(v/t) - \frac{1}{2}\delta \right)$$

$$= t g_k^\top v + \frac{1}{2} v^\top H_k v - \frac{1}{2}\delta t^2$$

$$= \begin{bmatrix} v \\ t \end{bmatrix}^\top \begin{bmatrix} H_k & g_k \\ g_k^\top & -\delta \end{bmatrix} \begin{bmatrix} v \\ t \end{bmatrix}.$$

Notice that (A.0.1) can be regarded as a problem of finding the leftmost eigenvector of a matrix. Hence, the randomized Lanczos algorithm (Kuczyński & Woźniakowski, 1992) can be utilized to solve (A.0.1). The solution of (A.0.1) is denoted by $[v_k; t_k]$. Using this solution, HSODM calculates the direction as follows:

$$d_k^{\mathrm{F}} = \begin{cases} v_k/t_k, & \text{if } |t_k| > \nu \\ \mathrm{sign}(-g_k^\top v_k)v_k, & \text{if } |t_k| \leq \nu \end{cases},$$

where $\nu \in (0, 1/2)$ is an arbitrary parameter.

Then, the algorithm updates the iterates according to the following rule:

$$x_{k+1} = \begin{cases} x_k + \eta_k d_k, & \text{if } \left\| d_k^{\mathrm{F}} \right\| > \Delta \\ x_k + d_k, & \text{if } \left\| d_k^{\mathrm{F}} \right\| \leq \Delta \end{cases}.$$

Here, $\eta_k$ denotes the step size, and $\Delta \in [0, \sqrt{2}/2]$ is a parameter set to the appropriate value to achieve the desired accuracy. If the first condition $\|d_k^{\mathrm{F}}\| > \Delta$ is met, the step size can be either set to $\eta_k = \Delta/\|d_k^{\mathrm{F}}\|$ or determined by a line search. If the second condition $\|d_k^{\mathrm{F}}\| > \Delta$ is satisfied, this algorithm can either terminate and output $x_{k+1}$, thereby obtaining an $\varepsilon$–SOSP, or reset both $\delta$ and $\nu$ to 0 and continue, thereby achieving local quadratic convergence. These steps are shown in Algorithm 2.

## B    Preparation of the theoretical analysis

### B.1    Existing lemmas

We introduce two important properties of random projections. The first property is that random projections approximately preserve norms with high probability. Formally, the following Johnson-Lindenstrauss (JL) lemma holds:

---

**Algorithm 2** Homogeneous Second-Order Descent Method (HSODM) (Zhang et al., 2022)

---

1: **function** HSODM$(n, \delta, \Delta, \nu, \text{max\_iter})$
2:     **for** $k = 1, \ldots, \text{max\_iter}$ **do**
3:         $(t_k, v_k) \leftarrow \text{solve\_subproblem}(g_k, H_k, \delta)$
4:         $d_k \leftarrow \begin{cases} v_k/t_k, & \text{if } |t_k| > \nu \\ \text{sign}(-\tilde{g}_k^\top \tilde{v}_k)v_k, & \text{otherwise} \end{cases}$
5:         **if** $\|d_k\| > \Delta$ **then**
6:             $\eta_k \leftarrow \Delta/\|d_k\|$                  ▷ or get from backtracking line search
7:             $x_{k+1} \leftarrow x_k + \eta_k d_k$
8:         **else**
9:             $x_{k+1} \leftarrow x_k + d_k$
10:             **terminate**         ▷ or continue with $(\delta, \nu) \leftarrow (0, 0)$ for local convergence

---

**Lemma B.1.** *[Lemma 5.3.2 in Vershynin (2018)] Let $P \in \mathbb{R}^{s \times n}$ be a random Gaussian matrix and $C$ be an absolute constant. Then, for any $x \in \mathbb{R}^n$ and any $\xi \in (0, 1)$, the following inequality holds with probability at least $1 - 2\exp\left(-C\xi^2 s\right)$:*

$$(1 - \xi)\|x\| \leq \|Px\| \leq (1 + \xi)\|x\|.$$

The second property is described by the following lemma, which states that $P^\top$ is an approximate isometry with high probability.

**Lemma B.2** (Theorem 4.6.1, Exercie 4.6.2, 4.6.3 in Vershynin (2018))**.** *Let $P \in \mathbb{R}^{s \times n}$ be a random Gaussian matrix and $\mathcal{C}$ be an absolute constant. The following inequality holds with probability at least $1 - 2\exp(-s)$:*

$$\forall y \in \mathbb{R}^s, \quad \left(\sqrt{\frac{n}{s}} - \mathcal{C}\right)\|y\| \leq \|P^\top y\| \leq \left(\sqrt{\frac{n}{s}} + \mathcal{C}\right)\|y\|.$$

We recall the following lemma from Nesterov (2018).

**Lemma B.3.** *Suppose Assumption 1 holds. Then for any $x, y \in \mathbb{R}^n$, we have*

$$|f(y) - f(x) - \nabla f(x)^\top (y - x)| \leq \frac{L}{2}\|y - x\|^2,$$

$$\|\nabla f(y) - \nabla f(x) - \nabla^2 f(x)(y - x)\| \leq \frac{M}{2}\|y - x\|^2,$$

$$\left| f(y) - f(x) - \nabla f(x)^\top (y - x) - \frac{1}{2}(y - x)^\top \nabla^2 f(x)(y - x) \right| \leq \frac{M}{6}\|y - x\|^3. \qquad \text{(B.1.1)}$$

B.2   THE OPTIMALITY CONDITIONS OF THE DIMENSION-REDUCED SUBPROBLEM

We recall the fundamental property in probability theory. For any events $E_1$ and $E_2$, we have

$$\Pr[E_1 \cap E_2] \geq 1 - (1 - \Pr[E_1]) - (1 - \Pr[E_2]).$$

This is used throughout this paper without further explicit mention. The optimality conditions of the dimension-reduced subproblem (2.2.1) are given by the following statements: there exists a non-negative random variable $\theta_k$ such that

$$\left(\tilde{F}_k + \theta_k I\right)\begin{bmatrix} \tilde{v}_k \\ t_k \end{bmatrix} = 0, \qquad (\text{B.2.1})$$

$$\tilde{F}_k + \theta_k I \succeq 0, \qquad (\text{B.2.2})$$

$$\theta_k \left(\|[\tilde{v}_k; t_k]\| - 1\right) = 0, \qquad (\text{B.2.3})$$

where

$$\tilde{F}_k = \begin{bmatrix} \tilde{H}_k & \tilde{g}_k \\ \tilde{g}_k^\top & -\delta \end{bmatrix}.$$

It immediately follows from (B.2.2) that

$$\lambda_{\min}(\tilde{F}_k) \geq -\theta_k. \tag{B.2.4}$$

We also obtain from (B.2.2)

$$\delta \leq \theta_k, \tag{B.2.5}$$

by considering the direction $[0, \cdots, 0, 1]^\top$.

We directly deduce the following result from (B.2.1).

**Corollary B.1.** (B.2.1) *implies the following equations*

$$\left(\tilde{H}_k + \theta_k I\right) \tilde{v}_k = -t_k \tilde{g}_k, \tag{B.2.6}$$

$$\tilde{g}_k^\top \tilde{v}_k = t_k \left(\delta - \theta_k\right).$$

*Furthermore, if $t_k = 0$, then*

$$\left(\tilde{H}_k + \theta_k I\right) \tilde{v}_k = 0, \tag{B.2.7}$$

$$\tilde{g}_k^\top \tilde{v}_k = 0$$

*hold. If $t_k \neq 0$, then*

$$\tilde{g}_k^\top \frac{\tilde{v}_k}{t_k} = \delta - \theta_k, \tag{B.2.8}$$

$$\left(\tilde{H}_k + \theta_k I\right) \frac{\tilde{v}_k}{t_k} = -\tilde{g}_k \tag{B.2.9}$$

*hold.*

We also obtain a slightly stronger inequality $\delta < \theta_k$ under an additional condition.

**Lemma B.4.** *If $g_k \neq 0$, then $\delta < \theta_k$ holds with probability 1.*

*Proof.* We first prove the following inequality:

$$\lambda_{\min}(\tilde{F}_k) < -\delta. \tag{B.2.10}$$

To prove this inequality, it suffices to show that $\tilde{F}_k + \delta I$ has negative curvature. Let us define

$$f(\eta, t) = \begin{bmatrix} -\eta \tilde{g}_k \\ t \end{bmatrix}^\top \left(\tilde{F}_k + \delta I\right) \begin{bmatrix} -\eta \tilde{g}_k \\ t \end{bmatrix}$$

$$= \eta^2 \tilde{g}_k^\top (\tilde{H}_k + \delta I) \tilde{g}_k - 2\eta t \|\tilde{g}_k\|^2.$$

Then for any fixed $t > 0$, we have

$$f(0, t) = 0, \quad \frac{\partial f(0, t)}{\partial \eta} = -2t \|\tilde{g}_k\|^2.$$

Since $g_k \neq 0$, we have $\|\tilde{g}_k\| \neq 0$ with probability 1, which implies $\frac{\partial f(0,t)}{\partial \eta} < 0$. Thus, for sufficiently small $\eta > 0$, we have that $f(\eta, t) < 0$. This shows that $\tilde{F}_k + \delta I$ has negative curvature. Finally, by combining (B.2.4) and (B.2.10), the proof is completed. $\square$

## C  PURE RANDOM SUBSPACE VARIANT OF HSODM

Unlike the pure random subspace variant of HSODM[6], RSHTR excludes the parameter $\nu$ present in HSODM. This exclusion is because our theoretical analysis in the fixed-radius strategy, which we focus on, does not depend on $\nu$. Therefore, we discussed it under $\nu \to +0$ for clarity. This section discusses that similar theoretical guarantees can be provided for any $\nu \in (0, 1/2)$ as in the case of $\nu \to +0$.

---

[6]Here, we cite the algorithm used in Zhang et al. (2022). Notice that in the newest version, Zhang et al. (2024), the stopping criterion is written differently, but it is equivalent to the one in Zhang et al. (2022).

---

**Algorithm 3** Pure random subspace variant of HSODM

---

1: **function** RSHTR$(s, n, \delta, \Delta, \nu, \text{max\_iter})$
2:  $\quad$ `global_mode = True`
3:  $\quad$ **for** $k = 1, \dots, \text{max\_iter}$ **do**
4:  $\quad\quad$ $P_k \leftarrow s \times n$ random Gaussian matrix with each element being from $\mathcal{N}(0, 1/s)$
5:  $\quad\quad$ $\tilde{g}_k \leftarrow P_k g_k$
6:  $\quad\quad$ $(t_k, \tilde{v}_k) \leftarrow$ optimal solution of (2.2.1) by eigenvalue computation
7:  $\quad\quad$ $(\tilde{g}_k, \tilde{H}_k, \delta)$
8:  $\quad\quad$ $d_k \leftarrow \begin{cases} P_k^\top \tilde{v}_k / t_k, & \text{if } |t_k| > \nu \\ \text{sign}(-\tilde{g}_k^\top \tilde{v}_k) P_k^\top \tilde{v}_k, & \text{otherwise} \end{cases}$
9:  $\quad\quad$ **if** $\|d_k\| > \Delta$ **then**
10: $\quad\quad\quad$ $\eta_k \leftarrow \Delta / \|d_k\|$ $\qquad\qquad\qquad\qquad$ ▷ or get from backtracking line search
11: $\quad\quad\quad$ $x_{k+1} \leftarrow x_k + \eta_k d_k$
12: $\quad\quad$ **else**
13: $\quad\quad\quad$ $x_{k+1} \leftarrow x_k + d_k$
14: $\quad\quad\quad$ **terminate** $\quad$ ▷ or continue with $(\delta, \nu, \texttt{global\_mode}) \leftarrow (0, 0, \texttt{False})$ for local conv.

---

## C.1 Analysis on fixed radius strategy

Here, we consider the case of a fixed step size, $\Delta$. When $|t_k| < \nu$, $d_k$ is given by $d_k = P_k^\top \tilde{v}_k$.

**Lemma C.1.** *Suppose that Assumption 1 holds. Let $\nu \in (0, 1/2), d_k = P_k^\top \tilde{v}_k$ and $\eta_k = \Delta / \|d_k\|$. If $|t_k| < \nu, g_k \neq 0$ and $\|d_k\| > \Delta$ , then we have*

$$f(x_{k+1}) - f(x_k) \leq -\frac{1}{2(\sqrt{n/s} + \mathcal{C})^2} \Delta^2 \delta + \frac{M}{6} \Delta^3$$

*with probability at least $1 - 2\exp(-s)$.*

*Proof.* By Corollary B.1, we have

$$\tilde{v}_k^\top \tilde{H}_k \tilde{v}_k = -\theta_k \|\tilde{v}_k\|^2 - t_k \tilde{v}_k^\top \tilde{g}_k, \tag{C.1.1}$$

$$\tilde{v}_k^\top \tilde{g}_k = t_k(\delta - \theta_k). \tag{C.1.2}$$

Since we have $\delta < \theta_k$ with probability 1 from Lemma B.4, it follows that

$$\text{sign}(-\tilde{g}_k^\top \tilde{v}_k) = \text{sign}(t_k) \tag{C.1.3}$$

with probability 1. Therefore we obtain

$$\begin{aligned} d_k^\top H_k d_k &= \tilde{v}_k^\top \tilde{H}_k \tilde{v}_k \quad \text{(by definition of } d_k) \\ &= -\theta_k \|\tilde{v}_k\|^2 - t_k \tilde{v}_k^\top \tilde{g}_k \quad \text{(by (C.1.1))} \\ &= -\theta_k \|\tilde{v}_k\|^2 - t_k^2(\delta - \theta) \quad \text{(by (C.1.2))}, \end{aligned} \tag{C.1.4}$$

$$\begin{aligned} g_k^\top d_k &= \text{sign}(-\tilde{g}_k^\top \tilde{v}_k) \tilde{g}_k^\top \tilde{v}_k \quad \text{(by definition of } d_k) \\ &= \text{sign}(-\tilde{g}_k^\top \tilde{v}_k) t_k(\delta - \theta_k) \quad \text{(by (C.1.2))} \\ &= |t_k|(\delta - \theta_k) \quad \text{(by (C.1.3))}. \end{aligned} \tag{C.1.5}$$

Since $\|d_k\| > \Delta$, it follows that $\eta_k = \Delta / \|d_k\| \in (0, 1)$. Thus, we have $\eta_k - \eta_k^2/2 \geq 0$. Hence

$$\left(\eta_k - \frac{\eta_k^2}{2}\right)(\delta - \theta_k) \leq 0. \tag{C.1.6}$$

---

**Algorithm 4** Backtracking Line Search

---

1: **function** BACKTRACKLINESEARCH($x_k, d_k, \gamma > 0, \beta \in (0,1)$)
2:     $\eta_k = 1$
3:     **for** $j_k = 0, 1, \ldots$ **do**
4:         **if** $f(x_k + \eta_k d_k) - f(x_k) \leq -\gamma \eta_k \|d_k\|^3/6$ **then**
5:             **return** $\eta_k$
6:         **else**
7:             $\eta_k \leftarrow \beta \eta_k$

---

Hence, the following inequality holds with probability at least $1 - 2\exp(-s)$.

$$f(x_{k+1}) - f(x_k) = f(x_k + \eta_k d_k) - f(x_k)$$

$$\leq \eta_k g_k^\top d_k + \frac{\eta_k^2}{2} d_k^\top H_k d_k + \frac{M}{6} \eta_k^3 \|d_k\|^3 \quad \text{(by (B.1.1))}$$

$$= \eta_k |t_k|(\delta - \theta_k) - \frac{\eta_k^2}{2}\theta_k \|\tilde{v}_k\|^2 - \frac{\eta_k^2}{2} t_k^2 (\delta - \theta_k) + \frac{M}{6} \eta_k^3 \|d_k\|^3 \quad \text{(by (C.1.4) and (C.1.5))}$$

$$\leq \eta_k t_k^2 (\delta - \theta_k) - \frac{\eta_k^2}{2}\theta_k \|\tilde{v}_k\|^2 - \frac{\eta_k^2}{2} t_k^2 (\delta - \theta_k) + \frac{M}{6} \eta_k^3 \|d_k\|^3 \quad \text{(by } 0 \leq |t_k| \leq 1\text{)}$$

$$= \left(\eta_k - \frac{\eta_k^2}{2}\right) t_k^2 (\delta - \theta_k) - \frac{\eta_k^2}{2}\theta_k \|\tilde{v}_k\|^2 + \frac{M}{6} \eta_k^3 \|d_k\|^3$$

$$\leq -\frac{\eta_k^2}{2}\theta_k \|\tilde{v}_k\|^2 + \frac{M}{6} \eta_k^3 \|d_k\|^3 \quad \text{(by (C.1.6))}$$

$$\leq -\frac{\eta_k^2}{2}\theta_k \cdot \frac{1}{(\sqrt{n/s} + \mathcal{C})^2} \|d_k\|^2 + \frac{M}{6} \eta_k^3 \|d_k\|^3 \quad \text{(by Lemma B.2)}$$

$$\leq -\frac{1}{2(\sqrt{n/s} + \mathcal{C})^2} \Delta^2 \delta + \frac{M}{6}\Delta^3 \quad \text{(by (B.2.5) and } \eta_k = \Delta/\|d_k\|\text{)}.$$

This completes the proof. $\qquad\qquad\qquad\qquad\qquad\qquad\qquad\qquad\qquad\qquad\qquad\quad$ $\square$

We now consider the case where $|t_k| \geq \nu$. In this case, $d_k$ is given by $d_k = P_k^\top \tilde{v}_k/t_k$. Since $|t_k| \geq \nu$ implies $t_k \neq 0$, we have the following result by using the same argument as in the proof of Lemma D.2.

**Lemma C.2.** *Suppose Assumption 1 holds. Let* $d_k = P_k^\top \tilde{v}_k/t_k$. *If* $|t_k| \geq \nu, \|d_k\| > \Delta$ *and* $\eta_k = \Delta/\|d_k\|$, *then*

$$f(x_{k+1}) - f(x_k) \leq -\frac{1}{2(\sqrt{n/s} + \mathcal{C})^2}\Delta^2 \delta + \frac{M}{6}\Delta^3$$

*holds with probability at least* $1 - 2\exp(-s)$.

By Lemmas C.1 and C.2, we can state that, when $\|d_k\| > \Delta$ holds, the objective function value decreases by at least $\frac{1}{2(\sqrt{n/s}+\mathcal{C})^2}\Delta^2\delta - \frac{M}{6}\Delta^3$ at each iteration, with probability at least $1 - 2\exp(-s)$.

## C.2 ANALYSIS CONSIDERING A LINE SEARCH STRATEGY

In this subsection, we analyze the case where the step size is selected by the line search algorithm shown in Algorithm 4. Specifically, we show that Algorithm 4 guarantees a sufficient decrease in the function value, similar to the fixed step size case. As in the previous subsection, we consider the cases $|t_k| < \nu$ and $|t_k| \geq \nu$ separately, providing a guarantee of function value decrease for each case (Lemma C.3 and Lemma C.4, respectively).

**Lemma C.3.** *Suppose that Assumption 1 holds. Let* $\nu \in (0, 1/2), |t_k| < \nu, d_k = \text{sign}(-\tilde{g}_k^\top \tilde{v}_k)P_k^\top \tilde{v}_k, \beta \in (0,1), \gamma > 0,$ *and* $\eta_k$ *be chosen by Algorithm 4. Then, with probability*

*at least $1 - 2\exp(-s)$, the number of iterations of Algorithm 4 is bounded above by*

$$\left\lceil \log_\beta \left( \frac{3\delta}{M+\gamma} \left( \sqrt{\frac{n}{s}} + \mathcal{C} \right)^{-3} \right) \right\rceil,$$

*and the decrease in the function value is bounded as follows:*

$$f(x_{k+1}) - f(x_k) \leq -\min\left\{ \frac{\sqrt{3}\gamma}{16} \left( \sqrt{\frac{n}{s}} - \mathcal{C} \right)^3, \frac{9\gamma\beta^3\delta^3}{2(M+\gamma)^3(\sqrt{n/s}+\mathcal{C})^6} \right\}.$$

*Proof.* Let $j_k$ be the number of iterations at which Algorithm 4 stops in the $k$-th outer iteration. If the line search terminates with $j_k = 0$, i.e., $\eta_k = 1$, then

$$
\begin{aligned}
f(x_k + \eta_k d_k) - f(x_k) &\leq -\frac{\gamma}{6}\eta_k^3 \|d_k\|^3 \\
&\leq -\frac{\gamma}{6} \|P_k^\top \tilde{v}_k\|^3 \\
&\leq -\frac{\gamma}{6} \left( \sqrt{\frac{n}{s}} - \mathcal{C} \right)^3 \|\tilde{v}_k\|^3 \quad \text{(by Lemma B.2)} \\
&\leq -\frac{\sqrt{3}\gamma}{16} \left( \sqrt{\frac{n}{s}} - \mathcal{C} \right)^3 \quad \text{(since } \|\tilde{v}_k\| = \sqrt{1 - |t_k|^2} \geq \sqrt{1 - \nu^2} = \sqrt{3}/2).
\end{aligned}
$$

Let us consider the case where Algorithm 4 does not stop at the $j$-th ($j \geq 0$) iteration. Then $f(x_k + \eta_k d_k) - f(x_k) > -\frac{\gamma}{6}\eta_k^3\|d_k\|^3$. Following the proof of Lemma C.1, we have

$$
\begin{aligned}
-\frac{\gamma}{6}\eta_k^3 \|d_k\|^3 &< f(x_k + \eta_k d_k) - f(x_k) \\
&\leq -\frac{\eta_k^2}{2}\theta_k \left( \sqrt{\frac{n}{s}} + \mathcal{C} \right)^{-2} \|d_k\|^2 + \frac{M}{6}\eta_k^3\|d_k\|^3 \\
&\leq -\frac{\eta_k^2}{2}\delta \left( \sqrt{\frac{n}{s}} + \mathcal{C} \right)^{-2} \|d_k\|^2 + \frac{M}{6}\eta_k^3\|d_k\|^3 \quad \text{(by (B.2.5))}.
\end{aligned}
$$

This implies $\eta_k > \frac{3\delta}{(M+\gamma)\|d_k\|} \left( \sqrt{\frac{n}{s}} + \mathcal{C} \right)^{-2}$ and $j < \log_\beta \left( \frac{3\delta}{(M+\gamma)\|d_k\|} \left( \sqrt{\frac{n}{s}} + \mathcal{C} \right)^{-2} \right)$. Since $\|d_k\| = \|P_k^\top \tilde{v}_k\| \leq \left( \sqrt{\frac{n}{s}} + \mathcal{C} \right) \|\tilde{v}_k\| \leq \left( \sqrt{\frac{n}{s}} + \mathcal{C} \right)$ due to Lemma B.2 and $\|\tilde{v}_k\| \leq 1$, the number $j_k$ of iterations where Algorithm 4 stops is bounded above by $\left\lceil \log_\beta \left( \frac{3\delta}{M+\gamma} \left( \sqrt{\frac{n}{s}} + \mathcal{C} \right)^{-3} \right) \right\rceil$. Moreover, the decrease in the function value can be bounded as follows:

$$
\begin{aligned}
f(x_k + \eta_k d_k) - f(x_k) &\leq -\frac{\gamma}{6}\eta_k^3\|d_k\|^3 \\
&= -\frac{\gamma}{6}\beta^{3j_k}\|d_k\|^3 \\
&\leq -\frac{9\gamma\beta^3\delta^3}{2(M+\gamma)^3(\sqrt{n/s}+\mathcal{C})^6},
\end{aligned}
$$

where the last inequality follows from $\beta^{j_k-1} \geq \frac{3\delta}{(M+\gamma)\|d_k\|} \left( \sqrt{\frac{n}{s}} + \mathcal{C} \right)^{-2}$. Note that since we used only Lemma B.2 as a probabilistic result, the probability that this proof holds is at least $1 - 2\exp(-s)$.

$\square$

**Lemma C.4.** *Suppose Assumption 1 holds. Let $\nu \in (0, 1/2)$, $|t_k| \geq \nu$, $d_k = P_k^\top \tilde{v}_k/t_k$, $\|d_k\| \geq \Delta$, $\beta \in (0, 1)$, $\gamma > 0$, and $\eta_k$ be chosen by Algorithm 4. Then, with probability at least $1 - 2\exp(-s)$, the number of iterations of Algorithm 4 is bounded above by*

$$\left\lceil \log_\beta \left( \frac{3\delta\nu}{M+\gamma} \left( \sqrt{\frac{n}{s}} + \mathcal{C} \right)^{-3} \right) \right\rceil,$$

*and the decrease in the function value is bounded as follows:*

$$f(x_{k+1}) - f(x_k) \leq -\min\left\{\frac{\gamma\Delta^3}{6}, \frac{9\gamma\beta^3\delta^3}{2(M+\gamma)^3(\sqrt{n/s}+\mathcal{C})^6}\right\}.$$

*Proof.* Let $j_k$ be the number of iterations at which Algorithm 4 stops in the $k$-th outer iteration. If the line search terminates with $j_k = 0$, i.e., $\eta_k = 1$, then

$$f(x_k + \eta_k d_k) - f(x_k) \leq -\frac{\gamma}{6}\eta_k^3\|d_k\|^3$$

$$\leq -\frac{\gamma}{6}\Delta^3 \quad \text{(since } \|d_k\| \geq \Delta\text{).}$$

Let us consider the case where Algorithm 4 does not stop at the $j$-th ($j \geq 0$) iteration. Then $f(x_k + \eta_k d_k) - f(x_k) > -\frac{\gamma}{6}\eta_k^3\|d_k\|^3$. Following the proof of Lemma C.2, we have

$$-\frac{\gamma}{6}\eta_k^3\|d_k\|^3 < f(x_k + \eta_k d_k) - f(x_k)$$

$$\leq -\frac{\eta_k^2}{2}\theta_k\left(\sqrt{\frac{n}{s}}+\mathcal{C}\right)^{-2}\|d_k\|^2 + \frac{M}{6}\eta_k^3\|d_k\|^3$$

$$\leq -\frac{\eta_k^2}{2}\delta\left(\sqrt{\frac{n}{s}}+\mathcal{C}\right)^{-2}\|d_k\|^2 + \frac{M}{6}\eta_k^3\|d_k\|^3 \quad \text{(by (B.2.5)).}$$

This implies $\eta_k > \frac{3\delta}{(M+\gamma)\|d_k\|}\left(\sqrt{\frac{n}{s}}+\mathcal{C}\right)^{-2}$ and $j < \log_\beta\left(\frac{3\delta}{(M+\gamma)\|d_k\|}\left(\sqrt{\frac{n}{s}}+\mathcal{C}\right)^{-2}\right)$. Therefore, $j_k$, the number of iterations where Algorithm 4 stops, is bounded above by $\left\lceil\log_\beta\left(\frac{3\delta\nu}{M+\gamma}\left(\sqrt{\frac{n}{s}}+\mathcal{C}\right)^{-3}\right)\right\rceil$. Here, we have used the fact that

$$\|d_k\| = \left\|P_k^\top\frac{\tilde{v}_k}{t_k}\right\|$$

$$\leq \left(\sqrt{\frac{n}{s}}+\mathcal{C}\right)\frac{\|\tilde{v}_k\|}{|t_k|} \quad \text{(by Lemma B.2)}$$

$$= \left(\sqrt{\frac{n}{s}}+\mathcal{C}\right)\frac{\sqrt{1-|t_k|^2}}{|t_k|}$$

$$\leq \left(\sqrt{\frac{n}{s}}+\mathcal{C}\right)\frac{1}{\nu} \quad \text{(since } |t_k| \geq \nu\text{).}$$

The decrease in the function value can be bounded as follows:

$$f(x_k + \eta_k d_k) - f(x_k) \leq -\frac{\gamma}{6}\eta_k^3\|d_k\|^3$$

$$= -\frac{9\gamma\beta^3\delta^3}{2(M+\gamma)^3(\sqrt{n/s}+\mathcal{C})^6},$$

where the last inequality follows from $\beta^{j_k-1} \geq \frac{3\delta}{(M+\gamma)\|d_k\|}\left(\sqrt{\frac{n}{s}}+\mathcal{C}\right)^{-2}$. Note that since we used only Lemma B.2 as a probabilistic result, the probability that this proof holds is at least $1 - 2\exp(-s)$. $\qquad\square$

From the above two lemmas (Lemmas C.3 and C.4), we can bound the number of line search iterations and the decrease in the function value for general $t_k$ as follows.

**Corollary C.1.** *Suppose Assumption 1 holds. Let $\nu \in (0, 1/2), \beta \in (0, 1), \gamma > 0, \Delta > 0$, and $\delta > 0$. Then, with probability at least $1 - 2\exp(-s)$, the number of linesearch iterations of Algorithm 4 is bounded above by*

$$\left\lceil\log_\beta\left(\frac{3\delta\nu}{M+\gamma}\left(\sqrt{\frac{n}{s}}+\mathcal{C}\right)^{-3}\right)\right\rceil,$$

*and the decrease in the function value is bounded as follows:*

$$f(x_{k+1}) - f(x_k) \leq -\min\left\{\frac{\sqrt{3}\gamma}{16}\left(\sqrt{\frac{n}{s}}-\mathcal{C}\right)^3, \frac{\gamma\Delta^3}{6}, \frac{9\gamma\beta^3\delta^3}{2(M+\gamma)^3(\sqrt{n/s}+\mathcal{C})^6}\right\}.$$

# D  Proofs for theoretical analysis

## D.1  Analysis of the case where $\|d_k\| > \Delta$

Let us consider the case where $\|d_k\| > \Delta$ and evaluate the amount of decrease of the objective function value at each iteration (Lemmas D.1 and D.2, leading to Lemma 3.1). Note that, under $\|d_k\| > \Delta$, the update rule is given by

$$x_{k+1} = x_k + \eta_k d_k,$$
$$\eta_k = \Delta / \|d_k\|.$$

First, we consider the case where $t_k = 0$. In this case, $d_k$ is given by $d_k = P_k^\top \tilde{v}_k$.

**Lemma D.1.** *Suppose that Assumption 1 holds. Let $d_k = P_k^\top \tilde{v}_k$ and $\eta_k = \Delta / \|d_k\|$. If $t_k = 0, g_k \neq 0$ and $\|d_k\| > \Delta$ , then we have*

$$f(x_{k+1}) - f(x_k) \leq -\frac{1}{2(\sqrt{n/s} + \mathcal{C})^2} \Delta^2 \delta + \frac{M}{6} \Delta^3$$

*with probability at least $1 - 2\exp(-s)$.*

*Proof.* By $t_k = 0$ and Corollary B.1, we have

$$\tilde{v}_k^\top \tilde{H}_k \tilde{v}_k = -\theta_k \|\tilde{v}_k\|^2, \tag{D.1.1}$$
$$\tilde{v}_k^\top \tilde{g}_k = 0. \tag{D.1.2}$$

Therefore we obtain

$$\begin{aligned}
d_k^\top H_k d_k &= \tilde{v}_k^\top \tilde{H}_k \tilde{v}_k \quad \text{(by definition of } d_k) \\
&= -\theta_k \|\tilde{v}_k\|^2 \quad \text{(by (D.1.1))}, \tag{D.1.3} \\
g_k^\top d_k &= \text{sign}(-\tilde{g}_k^\top \tilde{v}_k) \tilde{g}_k^\top \tilde{v}_k \quad \text{(by definition of } d_k) \\
&= 0 \quad \text{(by (D.1.2))}. \tag{D.1.4}
\end{aligned}$$

Hence, the following inequality holds with probability at least $1 - 2\exp(-s)$.

$$\begin{aligned}
f(x_{k+1}) - f(x_k) &= f(x_k + \eta_k d_k) - f(x_k) \\
&\leq \eta_k g_k^\top d_k + \frac{\eta_k^2}{2} d_k^\top H_k d_k + \frac{M}{6} \eta_k^3 \|d_k\|^3 \quad \text{(by (B.1.1))} \\
&= 0 - \frac{\eta_k^2}{2} \theta_k \|\tilde{v}_k\|^2 + \frac{M}{6} \eta_k^3 \|d_k\|^3 \quad \text{(by (D.1.3) and (D.1.4))} \\
&\leq -\frac{\eta_k^2}{2} \theta_k \cdot \frac{1}{(\sqrt{n/s} + \mathcal{C})^2} \|d_k\|^2 + \frac{M}{6} \eta_k^3 \|d_k\|^3 \quad \text{(by Lemma B.2)} \\
&\leq -\frac{1}{2(\sqrt{n/s} + \mathcal{C})^2} \Delta^2 \delta + \frac{M}{6} \Delta^3 \quad \text{(by (B.2.5) and } \eta_k = \Delta / \|d_k\|).
\end{aligned}$$

This ends the proof.  □

Next, we consider the case where $t_k \neq 0$. In this case, $d_k$ is given by $d_k = P_k^\top \tilde{v}_k / t_k$.

**Lemma D.2.** *Suppose Assumption 1 holds. Let $d_k = P_k^\top \tilde{v}_k / t_k$. If $t_k \neq 0, \|d_k\| > \Delta$ and $\eta_k = \Delta / \|d_k\|$, then*

$$f(x_{k+1}) - f(x_k) \leq -\frac{1}{2(\sqrt{n/s} + \mathcal{C})^2} \Delta^2 \delta + \frac{M}{6} \Delta^3$$

*holds with probability at least $1 - 2\exp(-s)$.*

*Proof.* Since $t_k \neq 0$, we obtain from (B.2.9)

$$\frac{\tilde{v}_k^\top}{t_k} \tilde{H}_k \frac{\tilde{v}_k}{t_k} = -\theta_k \frac{\|\tilde{v}_k\|^2}{t_k^2} - \frac{\tilde{g}_k^\top \tilde{v}_k}{t_k}. \tag{D.1.5}$$

Therefore, it follows that

$$\begin{aligned}
d_k^\top H_k d_k &= \frac{\tilde{v}_k^\top}{t_k} \tilde{H}_k \frac{\tilde{v}_k}{t_k} \\
&= -\theta_k \frac{\|\tilde{v}_k\|^2}{t_k^2} - \frac{\tilde{g}_k^\top \tilde{v}_k}{t_k} \quad \text{(by (D.1.5))} \\
&= -\theta_k \frac{\|\tilde{v}_k\|^2}{t_k^2} - g_k^\top d_k \quad \text{(by definition of } d_k), \tag{D.1.6}
\end{aligned}$$

$$\begin{aligned}
g_k^\top d_k &= \delta - \theta_k \quad \text{(by (B.2.8))} \\
&\leq 0 \quad \text{(by (B.2.5))}. \tag{D.1.7}
\end{aligned}$$

Since $\eta_k = \Delta/\|d_k\| \in (0,1)$, we have $\eta_k - \eta_k^2/2 \geq 0$. Thus we obtain

$$\left(\eta_k - \frac{\eta_k^2}{2}\right) g_k^\top d_k \leq 0 \quad \text{(by (D.1.7))}. \tag{D.1.8}$$

Therefore, the following inequality holds with probability at least $1 - 2\exp(-s)$.

$$\begin{aligned}
f(x_{k+1}) - f(x_k) &= f(x_k + \eta_k d_k) - f(x_k) \\
&\leq \eta_k g_k^\top d_k + \frac{\eta_k^2}{2} d_k^\top H_k d_k + \frac{M}{6} \eta_k^3 \|d_k\|^3 \quad \text{(by (B.1.1))} \\
&= \eta_k g_k^\top d_k + \frac{\eta_k^2}{2} \left(-\theta_k \frac{\|\tilde{v}_k\|^2}{t_k^2} - g_k^\top d_k\right) + \frac{M}{6} \eta_k^3 \|d_k\|^3 \quad \text{(by (D.1.6))} \\
&= \left(\eta_k - \frac{\eta_k^2}{2}\right) g_k^\top d_k - \frac{\eta_k^2}{2} \theta_k \frac{\|\tilde{v}_k\|^2}{t_k^2} + \frac{M}{6} \eta_k^3 \|d_k\|^3 \\
&\leq -\frac{\eta_k^2}{2} \theta_k \frac{\|\tilde{v}_k\|^2}{t_k^2} + \frac{M}{6} \eta_k^3 \|d_k\|^3 \quad \text{(by (D.1.8))} \\
&\leq -\frac{\eta_k^2}{2} \theta_k \cdot \frac{1}{(\sqrt{n/s} + \mathcal{C})^2} \|d_k\|^2 + \frac{M}{6} \eta_k^3 \|d_k\|^3 \quad \text{(by Lemma B.2)} \\
&\leq -\frac{1}{2(\sqrt{n/s} + \mathcal{C})^2} \Delta^2 \delta + \frac{M}{6} \Delta^3 \quad \text{(by (B.2.5) and } \eta_k = \Delta/\|d_k\|).
\end{aligned}$$

$\square$

By Lemmas D.1 and D.2, we can state that, when $\|d_k\| > \Delta$ holds, the objective function value decreases by at least $\frac{1}{2(\sqrt{n/s}+\mathcal{C})^2}\Delta^2\delta - \frac{M}{6}\Delta^3$ at each iteration with probability at least $1 - 2\exp(-s)$. This proves Lemma 3.1.

### D.2 ANALYSIS OF THE CASE WHERE $\|d_k\| \leq \Delta$

Now we consider the case where RSHTR satisfies $\|d_k\| \leq \Delta$ at the $k$-th iteration and outputs $x_k$. To investigate the property of $x_k$, we analyze $\|g_k\|$ and $\lambda_{\min}(\tilde{H}_k)$.

We first derive an upper and lower bound on the eigenvalues of $\nabla^2 f(x)$.

**Lemma D.3.** *Suppose Assumption 1 holds. Then for all $x \in \mathbb{R}^n$,*

$$\lambda_{\min}(\nabla^2 f(x)) \geq -L, \tag{D.2.1}$$
$$\lambda_{\max}(\nabla^2 f(x)) \leq L$$

*hold.*

*Proof.* Let $v$ be a unit eigenvector corresponding to the smallest eigenvalue. By definition of $\nabla^2 f(x)$, we have

$$\nabla^2 f(x)v = \lim_{h \to 0} \frac{\nabla f(x + hv) - \nabla f(x)}{h}.$$

To bound the right-hand side, we use the $L$-Lipschitz property. We obtain

$$\|\nabla^2 f(x)v\| \leq \lim_{h \to 0} \frac{\|\nabla f(x + hv) - \nabla f(x)\|}{|h|}$$

$$\leq \frac{L\|hv\|}{|h|} = L\|v\|.$$

Finally, since $\|\nabla^2 f(x)v\| = |\lambda_{\min}(\nabla^2 f(x))|\|v\|$, we obtain $\lambda_{\min}(\nabla^2 f(x)) \geq -L$. The proof of the second inequality is similar. $\qquad\square$

Next, we derive upper and lower bounds on the eigenvalues of $\tilde{H}_k$.

**Lemma D.4.** *Suppose Assumption 1 holds. Then we have*

$$\lambda_{\min}\left(\tilde{H}_k\right) \geq -\left(\sqrt{\frac{n}{s}} + C\right)^2 L, \tag{D.2.2}$$

$$\lambda_{\max}\left(\tilde{H}_k\right) \leq \left(\sqrt{\frac{n}{s}} + C\right)^2 L \tag{D.2.3}$$

*with probability at least $1 - 2\exp(-s)$.*

*Proof.* Let us consider the first inequality. Let $E = \left\{ \tilde{x} \in \mathbb{R}^s \mid \tilde{x}^\top \tilde{H}_k \tilde{x} < 0 \right\}$. Then we have

$$\lambda_{\min}\left(\tilde{H}_k\right) \geq \min\left\{0, \min_{\tilde{x} \in E} \frac{\tilde{x}^\top \tilde{H}_k \tilde{x}}{\|\tilde{x}\|^2}\right\}. \tag{D.2.4}$$

By Lemma B.2, we have

$$\|P_k^\top \tilde{x}\|^2 \leq \left(\sqrt{\frac{n}{s}} + C\right)^2 \|\tilde{x}\|^2 \tag{D.2.5}$$

with probability at least $1 - 2\exp(-s)$. Therefore, we obtain

$$\min\left\{0, \min_{\tilde{x} \in E} \frac{\tilde{x}^\top \tilde{H}_k \tilde{x}}{\|\tilde{x}\|^2}\right\} \geq \left(\sqrt{\frac{n}{s}} + C\right)^2 \min\left\{0, \min_{\tilde{x} \in E} \frac{\tilde{x}^\top P_k H_k P_k^\top \tilde{x}}{\|P_k^\top \tilde{x}\|^2}\right\} \quad \text{(by (D.2.5))}$$

$$\geq \left(\sqrt{\frac{n}{s}} + C\right)^2 \min\left\{0, \min_{x \in \mathbb{R}^n} \frac{x^\top H_k x}{\|x\|^2}\right\} \quad \text{(by } \mathrm{Im}(P_k^\top) \subset \mathbb{R}^n\text{)}$$

$$= \left(\sqrt{\frac{n}{s}} + C\right)^2 \min\left\{0, \lambda_{\min}\left(H_k\right)\right\}$$

$$\geq -\left(\sqrt{\frac{n}{s}} + C\right)^2 L \quad \text{(by (D.2.1)).}$$

Hence, by (D.2.4), the first inequality (D.2.2) holds with probability at least $1 - 2\exp(-s)$. The proof of (D.2.3) is similar. $\qquad\square$

Using this lemma, we can derive an upper bound on $\|g_k\|$.

**Proof of Lemma 3.2**

*Proof.* Notice that by Lemma B.1

$$P_k g_k = P_k H_k d_k + \theta_k \frac{\tilde{v}_k}{t_k}.$$

Hence, by Lemma B.1 and B.2 we deduce that with probability at least $1 - 2\exp\left(-\frac{C}{4}s\right) - 2\exp(-s)$,

$$\|g_k\| \leq 2L\sqrt{\frac{n}{s} + \mathcal{C}}\Delta + 2\frac{\theta_k}{\sqrt{\frac{n}{s} - \mathcal{C}}}\Delta.$$

Furthermore, by (B.2.9), we have that

$$\theta_k \leq \delta + \Delta\|g_k\|.$$

Therefore, we have

$$\left(1 - \frac{2\Delta^2}{\sqrt{n/s - \mathcal{C}}}\right)\|g_k\| \leq 2\Delta\left(L\sqrt{\frac{n}{s} + \mathcal{C}} + \frac{\Delta\delta}{\sqrt{n/s - \mathcal{C}}}\right).$$

Combining with $\Delta \leq \frac{1}{2\sqrt{2}}$, we deduce that

$$\frac{1}{2}\|g_k\| \leq 2\Delta\left(L\sqrt{\frac{n}{s} + \mathcal{C}} + \frac{\Delta\delta}{\sqrt{n/s - \mathcal{C}}}\right),$$

which completes the proof. $\qquad\square$

We can also derive a bound on $\lambda_{\min}(\tilde{H}_k)$.

**Proof of Lemma 3.3**

*Proof.* By (B.2.2) and Cauchy's interlace theorem, we have $\tilde{H}_k + \theta_k I \succeq 0$. With (D.4.3) and Lemma 3.2, it follows that

$$-\theta_k \geq -\left[8\Delta^2\left(\left(\sqrt{\frac{n}{s}} + \mathcal{C}\right)^2 L + \delta\right) + \delta\right]$$

with probability at least $1 - 2\exp\left(-\frac{C}{4}s\right) - 2\exp\left(-s\right)$. This implies

$$\tilde{H}_k \succeq -\left[8\Delta^2\left(\left(\sqrt{\frac{n}{s}} + \mathcal{C}\right)^2 L + \delta\right) + \delta\right]I.$$

$\qquad\square$

### D.3    PROOF OF THEOREM 3.1

*Proof.* Let us consider how many times we iterate the case where $|d_k| > \Delta$ at most. According to Lemma D.1 and Lemma D.2, the objective function decreases by at least

$$\frac{1}{2\left(\sqrt{n/s} + \mathcal{C}\right)^2}\Delta^2\delta - \frac{M}{6}\Delta^3 = \frac{\varepsilon^{3/2}}{3M^2}$$

with probability at least $1 - 2\exp\left(-s\right)$. Since the total amount of decrease does not exceed $D := f(x_0) - \inf_{x\in\mathbb{R}^n} f(x)$, the number of iterations for the case where $\|d_k\| > \Delta$ is at most $\lfloor 3M^2 D\varepsilon^{-3/2}\rfloor$ with probability at least $1 - 2\lfloor 3M^2 D\varepsilon^{-3/2}\rfloor \exp\left(-s\right)$. Also, since the algorithm terminates once it enters the case $\|d_k\| \leq \Delta$, the total number of iterations is at most $U_\varepsilon = \lfloor 3M^2 D\varepsilon^{-3/2}\rfloor + 1$ at least with the same probability as the above.

We can compute an $\varepsilon$–FOSP with probability at least $1 - 4\exp\left(-\frac{C}{4}s\right) - 4\exp\left(-s\right)$, which can be easily checked by applying Lemma 3.2 to the given $\delta$ and $\Delta$.

Therefore, RSHTR converges in $\lfloor 3M^2 D\varepsilon^{-3/2}\rfloor + 1 = O(\varepsilon^{-3/2})$ iterations with probability at least

$$1 - 2\lfloor 3M^2 D\varepsilon^{-3/2}\rfloor \exp\left(-s\right) - 4\exp\left(-\frac{C}{4}s\right) - 4\exp\left(-s\right)$$

$$\geq 1 - 4\exp\left(-\frac{C}{4}s\right) - (2U_\varepsilon + 2)\exp\left(-s\right).$$

$\qquad\square$

---

**Algorithm 5** RSHTR (modified)

---

1: **function** MODIFIED_RSHTR($s, n, \delta, \Delta, \text{max\_iter}$)
2:     global_mode = True
3:     **for** $k = 1, \ldots, \text{max\_iter}$ **do**
4:         $P_k \leftarrow s \times n$ random Gaussian matrix with each element being from $\mathcal{N}(0, 1/s)$
5:         $\tilde{g}_k \leftarrow P_k g_k$
6:         $(t_k, \tilde{v}_k) \leftarrow$ optimal solution of (2.2.1) by eigenvalue computation
7:         $d_k \leftarrow \begin{cases} P_k^\top \tilde{v}_k / t_k, & \text{if } t_k \neq 0 \\ P_k^\top \tilde{v}_k, & \text{otherwise} \end{cases}$
8:         **if** global_mode and $\|d_k\| > \Delta$ **then**
9:            $\eta_k \leftarrow \Delta / \|d_k\|$                           ▷ or get from backtracking line search
10:           $x_{k+1} \leftarrow x_k + \eta_k d_k$
11:         **else**
12:           $x_{k+1} \leftarrow x_k + d_k$
13:           **terminate** ▷ or continue with $(\delta, \text{global\_mode}) \leftarrow (0, \text{False})$ for local conv.

---

Notice that by Theorem 3.1 and Lemma 3.2, we obtain that

$$\|g_{k^*}\| \leq O\left( (n/s)^{(2+\tau)/2} \varepsilon \right),$$

where $k^*$ denotes the last iteration, $k$, where $\|d_k\| > \Delta$. Therefore, in order to obtain an $\varepsilon$-FOSP, we need to scale down $\varepsilon$ by $(n/s)^{(2+\tau)/2}$. We obtain therefore an iteration complexity of

$$O\left( \left( \left( \frac{n}{s} \right)^{-(2+\tau)/2} \varepsilon \right)^{-3/2} \right) = O\left( \left( \frac{n}{s} \right)^{(6+3\tau)/4} \varepsilon^{-3/2} \right).$$

This has $(n/s)^{(6+3\tau)/4}$ times the iteration complexity compared to a full-space algorithm. Especially, when $\tau$ is sufficiently small, it becomes almost $(n/s)^{3/2}$ times.

### D.4    IMPROVEMENT OF ITERATION COMPLEXITY UNDER THE ASSUMPTION OF LOW-EFFECTIVENESS

In this section, we show that the total complexity is improved under the assumption of low-effectiveness (Assumption 4) and the slight modification of the algorithm. The modified algorithm is shown in Algorithm 5. The difference from the original Algorithm 1 is that the algorithm updates $x$ before stopping after the algorithm satisfies the stopping criterion. In order to evaluate the gradient norm at the output point, we first introduce the following lemma.

**Lemma D.5.** *Assume that $f$ satisfies (3.4.1) and assume that $s \geq m = \text{rank}(\Pi)$. If $t_k \neq 0, d_k = P_k^\top \tilde{v}_k / t_k, \|d_k\| \leq \Delta$, then we have*

$$\|H_k d_k + g_k\| \leq \frac{1}{\zeta \left( 1 - \sqrt{\frac{m-1}{s}} \right)} \frac{\delta \Delta + \|g_k\| \Delta^2}{\sqrt{n/s} - \mathcal{C}}$$

*with probability at least $1 - 2\exp\left( -\frac{C}{4} s \right) - 2\exp\left( -s \right) - (\bar{C}\zeta)^{s-m+1} - e^{-\bar{c}s}$ for any $\zeta > 0$.*

*Proof.* Notice that since $f(x) = f(\Pi x)$ for all $x$, where $\Pi$ is a rank $m$ projection matrix, we have that there exists an orthogonal matrix $U \in \mathbb{R}^{n \times n}$ such that

$$H_k d_k + g_k = U \begin{pmatrix} r_k \\ 0 \end{pmatrix},$$

where $r_k \in \mathbb{R}^m$. Since $P_k$ has the same distribution as $P_k U$, we can assume that

$$P_k(H_k d_k + g_k) = P_k \begin{pmatrix} r_k \\ 0 \end{pmatrix} = \tilde{P}_k r_k,$$

where $\tilde{P}_k$ is an $s \times m$ random matrix whose elements are sampled independently from $\mathcal{N}(0, 1/s)$. By Lemma B.2, we deduce that

$$\sigma_{\min}(\tilde{P}_k)\|H_k d_k + g_k\| \leq \|P_k(H_k d_k + g_k)\|, \tag{D.4.1}$$

$$\|d_k\| = \frac{\|P_k^\top \tilde{v}_k\|}{|t_k|} \geq \left(\sqrt{\frac{n}{s}} - \mathcal{C}\right) \frac{\|\tilde{v}_k\|}{|t_k|} \tag{D.4.2}$$

with probability at least $1 - 2\exp\left(-\frac{C}{4}s\right) - 2\exp(-s)$. Moreover, by (B.2.8) in Corollary B.1, we deduce

$$\theta_k - \delta = -g_k^\top d_k \leq \|g_k\|\|d_k\| \leq \Delta\|g_k\|. \tag{D.4.3}$$

Hence,

$$\begin{aligned}
\|H_k d_k + g_k\| &\leq \frac{1}{\sigma_{\min}(\tilde{P}_k)}\|P_k(H_k d_k + g_k)\| \\
&= \frac{1}{\sigma_{\min}(\tilde{P}_k)}\theta_k\left\|\frac{\tilde{v}_k}{t_k}\right\| \quad \text{(by (B.2.9) in Corollary B.1)} \\
&\leq \frac{1}{\sigma_{\min}(\tilde{P}_k)}\frac{\theta_k}{\sqrt{n/s} - \mathcal{C}}\|d_k\| \quad \text{(by (D.4.2))} \\
&\leq \frac{1}{\sigma_{\min}(\tilde{P}_k)}\frac{\delta\Delta + \|g_k\|\Delta^2}{\sqrt{n/s} - \mathcal{C}} \quad \text{(by (D.4.3) and } \|d_k\| \leq \Delta).
\end{aligned}$$

Moreover, by (Rudelson & Vershynin, 2009, Theorem 1.1), we have

$$\forall \zeta > 0, \quad \Pr\left[\sigma_{\min}\left(\tilde{P}_k\right) \geq \zeta\left(1 - \sqrt{\frac{m-1}{s}}\right)\right] \geq 1 - (\bar{C}\zeta)^{s-m+1} - e^{-\bar{c}s}$$

for some constants $\bar{C}, \bar{c}$. Therefore, for any $\zeta > 0$, the following inequality holds with probability at least $1 - 2\exp\left(-\frac{C}{4}s\right) - 2\exp(-s) - (\bar{C}\zeta)^{s-m+1} - e^{-\bar{c}s}$.

$$\|H_k d_k + g_k\| \leq \frac{1}{\zeta\left(1 - \sqrt{\frac{m-1}{s}}\right)}\frac{\delta\Delta + \|g_k\|\Delta^2}{\sqrt{n/s} - \mathcal{C}}$$

$$\square$$

Here, we evaluate the gradient norm at the output point of RSHTR by using Lemma D.5.

**Lemma D.6.** *Suppose that Assumption 1 holds. Assume that $f$ satisfies (3.4.1) and assume that $s \geq m = rank(\Pi)$. If $\|d_k\| \leq \Delta \leq \frac{1}{2\sqrt{2}}$, then we have*

$$\|g_{k+1}\| \leq \frac{M}{2}\Delta^2 + \frac{1}{\sqrt{n/s} - \mathcal{C}}\left(2\delta\Delta + \frac{8\Delta^3}{\zeta\left(1 - \sqrt{\frac{m-1}{s}}\right)}\left(L\sqrt{n/s} + \mathcal{C} + \frac{\Delta\delta}{\sqrt{n/s} - \mathcal{C}}\right)\right)$$

*with probability of at least $1 - 4\exp\left(-\frac{C}{4}s\right) - 4\exp(-s) - (\bar{C}\zeta)^{s-m+1} - e^{-\bar{c}s}$ for any $\zeta > 0$.*

*Proof.* This is proved by the following inequalities.

$$\begin{aligned}
\|g_{k+1}\| &\leq \|g_{k+1} - H_k d_k - g_k\| + \|H_k d_k + g_k\| \\
&\leq \frac{M}{2}\|d_k\|^2 + \frac{\delta\Delta + \|g_k\|\Delta^2}{\sqrt{n/s} - \mathcal{C}} \quad \text{(by Lemma D.5)} \\
&\leq \frac{M}{2}\Delta^2 + \frac{1}{\sqrt{n/s} - \mathcal{C}}\left(\delta\Delta + \frac{4\Delta^3}{\zeta\left(1 - \sqrt{\frac{m-1}{s}}\right)}\left(L\sqrt{n/s} + \mathcal{C} + \frac{\Delta\delta}{\sqrt{n/s} - \mathcal{C}}\right)\right) \quad \text{(by Lemma 3.2)}.
\end{aligned}$$

Here, $\zeta$ is any positive number. The second and the third inequalities both hold with probability at least $1 - 2\exp\left(-\frac{C}{4}s\right) - 2\exp(-s) - (\bar{C}\zeta)^{s-m+1} - e^{-\bar{c}s}$. Therefore, this lemma holds with probability at least $1 - 4\exp\left(-\frac{C}{4}s\right) - 4\exp(-s) - (\bar{C}\zeta)^{s-m+1} - e^{-\bar{c}s}$. $\square$

To evaluate the iteration complexity using the same parameters as in Theorem 3.1, we substitute these parameters into Lemma D.6. This shows that Algorithm 5 achieves $\sqrt{n/s}\varepsilon$-FOSP in $O(\varepsilon^{-3/2})$ iterations. Therefore, by rescaling $\varepsilon$, it achieves $\varepsilon$-FOSP in $O((n/s)^{3/4}\varepsilon^{-3/2})$ iterations.

### D.5   $\varepsilon$–SOSP under the assumption of Shao (2022)

Let us introduce the following assumption.

**Assumption 5.** Let $\xi \in (0, 1)$, define $r = \operatorname{rank}(\nabla^2 f(x^*))$, $\lambda_1$ be the maximum non-zero eigenvalues of $\nabla^2 f(x^*)$, and $\lambda_r$ be the minimum non-zero eigenvalues of $\nabla^2 f(x^*)$. Then, the following inequality holds

$$1 - \xi + 16\frac{r-1}{s}\frac{1+\xi}{1-\xi}\frac{\lambda_1}{\lambda_r} \geq 0.$$

By the contraposition of (Shao, 2022, Lemma 5.6.6), under Assumption 5, the lower bound on the minimum eigenvalue of $H^*$ is proportional to the lower bound on the minimum eigenvalue of $P^*H^*P^{*\top}$ with high probability. Therefore, Lemma 3.3 leads to the following theorem.

**Theorem D.1** (Global convergence to an $\varepsilon$–SOSP under Assumption 5). *Suppose Assumptions 1 and 5 hold. Set $\varepsilon, \delta$ and $\Delta$ the same as Theorem 3.1, i.e.,*

$$0 < \varepsilon \leq \frac{M^2}{8}, \quad \delta = \left(\sqrt{\frac{n}{s}} + \mathcal{C}\right)^2 \sqrt{\varepsilon} \quad and \quad \Delta = \frac{\sqrt{\varepsilon}}{M}.$$

*If there exists a positive constant $\tau$ such that $s = O(n\varepsilon^{1/\tau})$, then RSHTR converges to an $\varepsilon$–SOSP in at most $O(\varepsilon^{-3/2})$ iterations with probability at least*

$$(0.9999)^{r-1}\left(1 - 2\exp\left(-\frac{s\xi^2}{C_3}\right)\right) - 6\exp\left(-\frac{C}{4}s\right) - (2U_\varepsilon + 2)\exp(-s),$$

*where $U_\varepsilon := \lfloor 3M^2 D\varepsilon^{-3/2}\rfloor + 1$.*

*Proof.* By Theorem 3.1, RSHTR converges to an $\varepsilon$–FOSP in at most $O(\varepsilon^{-3/2})$ iterations with probability at least,

$$1 - 4\exp\left(-\frac{C}{4}s\right) - (2U_\varepsilon + 2)\exp(-s).$$

Let $x^*$ denotes the $\varepsilon$–FOSP. We now proceed to prove that $x^*$ is also an $\varepsilon$–SOSP as well. By applying Lemma 3.3 to the given $\delta$ and $\Delta$, we have

$$\lambda_{\min}\left(P^*H^*P^{*\top}\right) \geq -\left(\frac{8\varepsilon}{M^2}\left(\sqrt{\frac{n}{s}} + \mathcal{C}\right)^2(L + \sqrt{\varepsilon}) + \left(\sqrt{\frac{n}{s}} + \mathcal{C}\right)^2\sqrt{\varepsilon}\right) \qquad \text{(D.5.1)}$$

with probability at least $1 - 2\exp\left(-\frac{C}{4}s\right) - 2\exp(-s)$. Hence, by using the contraposition of (Shao, 2022, Lemma 5.6.6) and denoting $\kappa_H = \min\{0, \lambda_1/\lambda_r\}$, we have

$$\lambda_{\min}(H^*) \geq -\left(1 - \xi + 16\frac{r-1}{s}\frac{1+\xi}{1-\xi}\kappa_H\right)^{-1}$$
$$\cdot\left(\frac{8\varepsilon}{M^2}\left(\sqrt{\frac{n}{s}} + \mathcal{C}\right)^2(L + \sqrt{\varepsilon}) + \left(\sqrt{\frac{n}{s}} + \mathcal{C}\right)^2\sqrt{\varepsilon}\right)$$
$$= \Omega(-\sqrt{\varepsilon})$$

with probability at least $(0.9999)^{r-1} \left( 1 - 2\exp\left( -\frac{s\xi^2}{C_3} \right) \right)$. This shows that $x^*$ is an $\varepsilon$–SOSP and the lower bound on the probability is given as follows:

$$(0.9999)^{r-1} \left( 1 - 2\exp\left( -\frac{s\xi^2}{C_3} \right) \right)$$

$$- \left( 4\exp\left( -\frac{C}{4}s \right) + (2U_\varepsilon + 2)\exp(-s) \right)$$

$$- \left( 2\exp\left( -\frac{C}{4}s \right) + 2\exp(-s) \right)$$

$$\geq (0.9999)^{r-1} \left( 1 - 2\exp\left( -\frac{s\xi^2}{C_3} \right) \right) - 6\exp\left( -\frac{C}{4}s \right) - (2U_\varepsilon + 4)\exp(-s).$$

$\square$

**Proof of Lemma 3.4**

*Proof.* $H^*$ can be expressed as $H^* = U^* D^* U^{*\top}$ using an orthogonal matrix $U^*$ and a diagonal matrix $D^*$. Here,

$$D^* = \mathrm{diag}(\lambda_1, \ldots, \lambda_r)$$

is the diagonal matrix with eigenvalues $\lambda_1, \ldots, \lambda_r$. Note that $\lambda_{r+1} = \cdots = \lambda_n = 0$. Hence, it follows that

$$P^* H^* P^{*\top} = P^* U^* D^* U^{*\top} P^{*\top}$$

$$= \hat{P}^* D^* \hat{P}^{*\top}$$

$$= \hat{P}_1^* D_1^* \hat{P}_1^{*\top}, \tag{D.5.2}$$

where $\hat{P}_1^*$ is the first $r$ columns of $\hat{P}^*$, and $D_1^*$ is the leading principal minor of order $r$ of $D^*$. Here, $\hat{P}^*$ is also a random Gaussian matrix due to the orthogonality of $U^*$. Therefore, $\hat{P}_1^*$ is full column rank with probability 1. This implies that

$$\forall y \in \mathbb{R}^r, \exists x \in \mathbb{R}^s \text{ s.t. } \hat{P}_1^{*\top} x = y \tag{D.5.3}$$

with probability 1. Hence, the following holds with probability 1.

$$\lambda_{\min}(H^*) = \min_{z \in \mathbb{R}^n} \frac{z^\top H^* z}{\|z\|^2}$$

$$= \min_{y \in \mathbb{R}^r} \frac{y^\top D_1^* y}{\|y\|^2}$$

$$\geq \min_{x \in \mathbb{R}^s} \frac{x^\top \hat{P}_1^* D_1^* \hat{P}_1^{*\top} x}{\|P_1^{*\top} x\|^2} \quad \text{(by (D.5.3))}$$

$$\geq \min \left\{ \min_{x \in E} \frac{x^\top \hat{P}_1^* D_1^* \hat{P}_1^{*\top} x}{\|P_1^{*\top} x\|^2}, 0 \right\}$$

$$\text{where } E := \left\{ x \in \mathbb{R}^s \mid x^\top \hat{P}_1^* D_1^* \hat{P}_1^{*\top} x < 0 \right\}$$

$$\geq \min \left\{ \min_{x \in E} \frac{1}{\sigma_{\min}\left(\hat{P}_1^*\right)^2} \frac{x^\top \hat{P}_1^* D_1^* \hat{P}_1^{*\top} x}{\|x\|^2}, 0 \right\}$$

$$\left( \text{by} \|\hat{P}_1^{*\top} x\|^2 \geq \sigma_{\min}\left(\hat{P}_1^*\right) \|x\| \right)$$

$$= \frac{1}{\sigma_{\min}\left(\hat{P}_1^*\right)^2} \min \left\{ \lambda_{\min}\left( \hat{P}_1^* D_1^* \hat{P}_1^{*\top} \right), 0 \right\}$$

$$= \frac{1}{\sigma_{\min}\left(\hat{P}_1^*\right)^2} \min \left\{ \lambda_{\min}\left( P^* H^* P^{*\top} \right), 0 \right\} \quad \text{(by (D.5.2))}.$$

Moreover, by (Rudelson & Vershynin, 2009, Theorem 1.1), we have

$$\forall \zeta > 0, \quad \Pr\left[\sigma_{\min}\left(\hat{P}_1^*\right) \geq \zeta\left(1 - \sqrt{\frac{r-1}{s}}\right)\right] \geq 1 - (\bar{C}\zeta)^{s-r+1} - e^{-\bar{c}s}$$

for some constants $\bar{C}, \bar{c}$. Therefore, for any $\zeta > 0$, the following inequality holds with probability at least $1 - (\bar{C}\zeta)^{s-r+1} - e^{-\bar{c}s}$.

$$\lambda_{\min}(H^*) \geq \frac{1}{\zeta^2\left(1 - \sqrt{\frac{r-1}{s}}\right)^2} \min\left\{\lambda_{\min}\left(P^* H^* P^{*\top}\right), 0\right\}.$$

$\square$

**Proof of Theorem 3.2**

*Proof.* By following the same argument as in the proof of Theorem D.1 up to (D.5.1), we obtain

$$\lambda_{\min}\left(P^* H^* P^{*\top}\right) \geq \Omega(-\sqrt{\varepsilon})$$

with probability at least $1 - 6\exp\left(-\frac{C}{4}s\right) - (2U_\varepsilon + 4)\exp\left(-s\right)$. Applying Lemma 3.4 with $\zeta = \bar{C}/e$, we have $\lambda_{\min}(H^*) \geq \Omega\left(-\sqrt{\varepsilon}\right)$ with probability at least

$$1 - 6\exp\left(-\frac{C}{4}s\right) - (2U_\varepsilon + 4)\exp\left(-s\right) - \exp\left(-s + r - 1\right) - \exp(-\bar{c}s)$$

for some constant $\bar{c}$. This completes the proof. $\square$

## D.6 Local convergence

We note that under Assumption 3, there exists $\mu > 0$ and $\bar{R} > 0$ such that

$$\forall x \in B(\bar{x}, \bar{R}), \ \nabla^2 f(x) \succeq \mu I, \tag{D.6.1}$$

where $B(x, R) := \{y \in \mathbb{R}^n \mid \|y - x\| \leq R\}$. Let us first discuss the special case, $x_k = \bar{x}$, which is equivalent to $g_k = 0$ by convexity from Assumption 3.

**Lemma D.7.** *Suppose that Assumption 3 holds. If $g_k = 0$, then $x_{k+1} = x_k$ with probability 1.*

Lemma D.7 states that the iterates do not move away from $\bar{x}$ once it is reached. Since staying at $\bar{x}$ achieves any local convergence rate, we ignore this case.

*Proof.* By applying (B.2.6) to $g_k = 0$, we have $(\tilde{H}_k + \theta_k I)\tilde{v}_k = 0$. This implies that $\tilde{v}_k = 0$ or $(-\theta_k, \tilde{v}_k)$ is an eigenpair of $\tilde{H}_k$.

We show that the latter case is impossible by supposing it and leading to a contradiction. Suppose that $(-\theta_k, \tilde{v}_k)$ is an eigenpair of $\tilde{H}_k$. This implies $\lambda_{\min}\left(\tilde{H}_k\right) \leq -\theta_k \leq 0$. Thus, we get

$$\exists y \in \mathbb{R}^s \setminus \{0\} \text{ s.t. } \frac{y^\top \tilde{H}_k y}{\|y\|^2} \leq 0.$$

Since $y \neq 0$, we have $0 < \|P_k^\top y\|$ with probability 1. Therefore, $\|y^2\|/\|P_k^\top y\|^2 > 0$ follows. Thus, by multiplying (D.6.2) by $\|y^2\|/\|P_k^\top y\|^2 > 0$, we obtain

$$\exists y \in \mathbb{R}^s \text{ s.t. } \frac{y^\top \tilde{H}_k y}{\|P_k^\top y\|^2} \leq 0.$$

By taking $z = P_k^\top y$, it follows that

$$\exists z \in \mathbb{R}^n \text{ s.t. } \frac{z^\top H_k z}{\|z\|^2} \leq 0.$$

Therefore $\lambda_{\min}(H_k) \le 0$ follows. However, this contradicts Assumption 3.

From the above argument, we have $\tilde{v}_k = 0$. Since $\tilde{v}_k = 0$ implies $t_k \ne 0$ from (B.2.3), $d_k$ is defined as

$$d_k = P_k^\top \tilde{v}_k / t_k = 0.$$

Therefore, $\|d_k\| \le \Delta$ holds and $x_{k+1} = x_k + d_k = x_k$ follows. $\qquad\square$

Next, we show that in a sufficiently small neighborhood of a local minimizer, we have $\|d_k\| \le \Delta$. To this end, we first present the following auxiliary lemma.

**Lemma D.8.** *Under Assumption 3, $t_k \ne 0$ with probability 1.*

*Proof.* Suppose on the contrary that $t_k = 0$, then $(-\theta_k, \tilde{v}_k)$ is an eigenpair of $\tilde{H}_k$ by (B.2.7) in Corollary B.1. Thus we have $\lambda_{\min}(\tilde{H}_k) \le -\theta_k < -\delta \le 0$ with probability 1 by $g_k \ne 0$ and Lemma B.4. This implies that

$$\exists y \in \mathbb{R}^s \text{ s.t. } \frac{y^\top \tilde{H}_k y}{\|y\|^2} < 0. \tag{D.6.2}$$

Note that $\|y\| \ne 0$ and $\|P_k^\top y\| \ne 0$ hold since the numerator and denominator of (D.6.2) are both non-zero. By multiplying (D.6.2) by $\|y\|^2 / \|P_k^\top y\|^2 > 0$, it follows that

$$\exists y \in \mathbb{R}^s \text{ s.t. } \frac{y^\top \tilde{H}_k y}{\|P_k^\top y\|^2} < 0.$$

By considering $z = P_k^\top y$, we obtain

$$\exists z \in \mathbb{R}^n \text{ s.t. } \frac{z^\top H_k z}{\|z\|^2} < 0.$$

Therefore $\lambda_{\min}(H_k) < 0$ follows. However, this contradicts Assumption 3. The proof is completed. $\qquad\square$

By Lemma D.8, under Assumption 3, we have $d_k = P_k^\top \frac{\tilde{v}_k}{t_k}$ with probability 1. This leads to the following lemma.

**Lemma D.9.** *Under Assumption 3, we have $\|d_k\| \le \Delta$ for sufficiently large $k$ with probability at least $1 - 2\exp\left(-\frac{C}{4}s\right) - 4\exp\left(-s\right)$.*

*Proof.* From Lemma D.8 and (B.2.9), we have the following with probability 1.

$$\frac{\tilde{v}_k}{t_k} = -(\tilde{H}_k + \theta_k I)^{-1} \tilde{g}_k.$$

Therefore, by multiplying $P_k^\top$ from the left, it follows that

$$d_k = P_k^\top \frac{\tilde{v}_k}{t_k} = -P_k^\top (\tilde{H}_k + \theta_k I)^{-1} \tilde{g}_k.$$

Thus, we obtain

$$\begin{aligned}
\|d_k\| &\le \|P_k^\top\| \|(\tilde{H}_k + \theta_k I)^{-1}\| \|\tilde{g}_k\| \\
&\le \|P_k^\top\| \|(\tilde{H}_k + \theta_k I)^{-1}\| \|g_k\|/2 \quad \text{(by Lemma B.1)} \\
&\le \frac{\sqrt{n/s} + \mathcal{C}}{2} \|(\tilde{H}_k + \theta_k I)^{-1}\| \|g_k\| \quad \text{(by Lemma B.2)}.
\end{aligned}$$

Here the second and third inequalities hold with probability at least $1 - 2\exp\left(-\frac{C}{4}s\right)$ and $1 - 2\exp\left(-s\right)$ respectively. By Lemma D.10 and (D.6.1), we have $\tilde{H}_k + \theta_k I \succeq \left(\sqrt{\frac{n}{s}} - \mathcal{C}\right)^2 \mu + \theta_k$

with probability at least $1 - 2\exp(-s)$. Hence, we have

$$
\begin{aligned}
\|d_k\| &\leq \frac{(\sqrt{n/s} + \mathcal{C})/2}{(\sqrt{n/s} - \mathcal{C})^2\mu + \theta_k}\|g_k\| \\
&\leq \frac{(\sqrt{n/s} + \mathcal{C})/2}{(\sqrt{n/s} - \mathcal{C})^2\mu}\|g_k\| \\
&\leq \frac{(\sqrt{n/s} + \mathcal{C})}{2(\sqrt{n/s} - \mathcal{C})^2}\frac{\|g_k\|}{\mu}.
\end{aligned}
$$

We have $\|g_k\| \to 0$ by Assumption 3, which leads to $\|d_k\| \leq \Delta$ for sufficiently large $k$. □

Here, we present an auxiliary lemma for the proof of Theorem 3.3.

**Lemma D.10.** *We have*

$$
\lambda_{\min}\left(\tilde{H}_k\right) \geq \left(\sqrt{\frac{n}{s}} - \mathcal{C}\right)^2 \lambda_{\min}(H_k),
$$

$$
\lambda_{\max}\left(\tilde{H}_k\right) \leq \left(\sqrt{\frac{n}{s}} + \mathcal{C}\right)^2 \lambda_{\max}(H_k)
$$

*with probability at least $1 - 2\exp(-s)$.*

*Proof.* It follows that for any $x \in \mathbb{R}^n$, we have $\frac{x^\top H_k x}{\|x\|^2} \geq \lambda_{\min}(H_k)$. Therefore, by setting $x = P_k^\top y$, we have $\frac{y^\top P_k H_k P_k^\top y}{\|P_k^\top y\|^2} \geq \lambda_{\min}(H_k)$ for any $y \in \mathbb{R}^s$. Using Lemma B.2, we have $\left(\sqrt{\frac{n}{s}} - \mathcal{C}\right)^2 \|y\|^2 \leq \|P_k^\top y\|^2$ with probability at least $1 - 2\exp(-s)$. This implies the following inequality holds.

$$
\frac{y^\top P_k H_k P_k^\top y}{\|y\|^2} \geq \left(\sqrt{\frac{n}{s}} - \mathcal{C}\right)^2 \lambda_{\min}(H_k).
$$

This inequality holds for any $y \in \mathbb{R}^s$, which proves the first equation. The proof of the second equation follows a similar argument. □

D.6.1  PROOF OF THEOREM 3.3

*Proof.*

$$
\begin{aligned}
\sqrt{\bar{H}}(x_{k+1} - \bar{x}) &= \sqrt{\bar{H}}(x_{k+1} - x_k) + \sqrt{\bar{H}}(x_k - \bar{x}) \\
&= \sqrt{\bar{H}}d_k + \sqrt{\bar{H}}(x_k - \bar{x}) \\
&= -\sqrt{\bar{H}}P_k^\top(\tilde{H}_k + \theta_k I)^{-1}P_k g_k + \sqrt{\bar{H}}(x_k - \bar{x}) \\
&= -\sqrt{\bar{H}}P_k^\top(\tilde{H}_k + \theta_k I)^{-1}P_k H_k(x_k - \bar{x}) \\
&\quad - \sqrt{\bar{H}}P_k^\top(\tilde{H}_k + \theta_k I)^{-1}P_k(g_k - H_k(x_k - \bar{x}_k)) \\
&\quad + \sqrt{\bar{H}}(x_k - \bar{x}) \\
&= -A - B + \sqrt{\bar{H}}(x_k - \bar{x}).
\end{aligned}
$$

Here, we define

$$
A := \sqrt{\bar{H}}P_k^\top(\tilde{H}_k + \theta_k I)^{-1}P_k H_k(x_k - \bar{x})
$$

and

$$
B := \sqrt{\bar{H}}P_k^\top(\tilde{H}_k + \theta_k I)^{-1}P_k(g_k - H_k(x_k - \bar{x}_k)).
$$

To bound $B$, we give a bound to $\|P_k^\top (P_k H_k P_k^\top + \theta_k I)^{-1} P_k\|$. By Lemma D.10, $P_k H_k P_k^\top$ is invertible with probability at least $1 - 2\exp(-s)$. Therefore, we have

$$\|P_k^\top (P_k H_k P_k^\top + \theta_k I)^{-1} P_k\| \leq \|P_k^\top (P_k H_k P_k^\top)^{-1} P_k\|. \tag{D.6.3}$$

Moreover, the right-hand side satisfies the following inequality.

$$\|P_k^\top (P_k H_k P_k^\top)^{-1} P_k\| \leq \|P_k^\top\|^2 \left( \left( \sqrt{\frac{n}{s}} - \mathcal{C} \right)^2 \lambda_{\min}(H_k) \right)^{-1} \quad \text{(by Lemma D.10)}$$

$$\leq \frac{(\sqrt{n/s} + \mathcal{C})^2}{(\sqrt{n/s} - \mathcal{C})^2} \frac{1}{\mu} \quad \text{(by Lemma B.2 and (D.6.1)).} \tag{D.6.4}$$

Here, the first and the second inequalities both hold with probability at least $1 - 2\exp(-s)$. Therefore, combining (D.6.3) and (D.6.4), we have

$$\|P_k^\top (P_k H_k P_k^\top + \theta_k I)^{-1} P_k\| \leq \frac{(\sqrt{n/s} + \mathcal{C})^2}{(\sqrt{n/s} - \mathcal{C})^2} \cdot \frac{1}{\mu} \tag{D.6.5}$$

with probability at least $1 - 6\exp(-s)$.

By Taylor expansion at $\bar{x}$ of $\nabla f$, we obtain $\|g_k - H_k(x_k - \bar{x})\| = O(\|x_k - \bar{x}\|^2)$. Combining this with (D.6.5), we get $B = O(\|x_k - \bar{x}\|^2)$ with probability at least $1 - 6\exp(-s)$.

Next, to bound $A$, we further decompose $A = A_1 + A_2$ such that

$$A_1 := \sqrt{\bar{H}} P_k^\top (P_k \bar{H} P_k^\top + \theta_k I)^{-1} P_k \bar{H}(x_k - \bar{x})$$

$$A_2 := \sqrt{\bar{H}} P_k^\top (P_k \bar{H} P_k^\top + \theta_k I)^{-1} P_k (H_k - \bar{H})(x_k - \bar{x}).$$

Since $\|H_k - \bar{H}\|$ tends to $0$ and (D.6.5), we have $\|A_2\| = o(\|x_k - \bar{x}\|)$. This leads to

$$\left\| \sqrt{\bar{H}}(x_{k+1} - \bar{x}) \right\| \leq \left\| -A_1 + \sqrt{\bar{H}}(x_k - \bar{x}) \right\| + o(\|x_k - \bar{x}\|),$$

which holds with probability at least $1 - 6\exp(-s)$. Therefore, it remains to bound $\left\| -A_1 + \sqrt{\bar{H}}(x_k - \bar{x}) \right\|$. This is further decomposed as $\|A_3 - A_4\|$, where

$$A_3 := \left( I - \sqrt{\bar{H}} P_k^\top (P_k \bar{H} P_k^\top)^{-1} P_k \sqrt{\bar{H}} \right) \sqrt{\bar{H}}(x_k - \bar{x}),$$

$$A_4 := \sqrt{\bar{H}} P_k^\top \left( (P_k \bar{H} P_k^\top + \theta_k I)^{-1} - (P_k \bar{H} P_k^\top)^{-1} \right) P_k \bar{H}(x_k - \bar{x}).$$

Here, $\|A_4\| = o(\|x_k - \bar{x}\|)$ holds for the following reason. By (B.2.8), Lemma D.8 and $\delta = 0$, we have $\theta_k = -g_k d_k$ with probability 1 and thus

$$\|\theta_k\| \leq \|g_k\| \|d_k\|$$
$$\leq \Delta \|g_k\| \quad \text{(by Lemma D.9)}$$

with probability at least $1 - 2\exp\left(-\frac{C}{4}s\right) - 4\exp(-s)$. This leads to $\theta_k \to 0$ since $\|g_k\|$ tends to 0. Therefore, $\left\| (P_k \bar{H} P_k^\top + \theta_k I)^{-1} - (P_k \bar{H} P_k^\top)^{-1} \right\|$ tends to 0, which implies $\|A_4\| = o(\|x_k - \bar{x}\|)$. Hence, it remains to bound $\|A_3\|$. Using the fact that $\sqrt{\bar{H}} P_k^\top (P_k \bar{H} P_k^\top)^{-1} P_k \sqrt{\bar{H}}$ is an orthogonal projection, this is bounded from above by

$$\sqrt{1 - \frac{\lambda_{\min}(\bar{H})}{2\lambda_{\max}(P_k \bar{H} P_k^\top)}} \left\| \sqrt{\bar{H}}(x_k - \bar{x}) \right\|.$$

Thus we obtain that

$$\left\| \sqrt{\bar{H}}(x_{k+1} - \bar{x}) \right\| \leq \sqrt{1 - \frac{\lambda_{\min}(\bar{H})}{2\lambda_{\max}(P_k \bar{H} P_k^\top)}} \left\| \sqrt{\bar{H}}(x_k - \bar{x}) \right\| + o(\|x_k - \bar{x}\|).$$

Therefore, when

$$\sqrt{1 - \frac{\lambda_{\min}(\bar{H})}{4\lambda_{\max}(P_k \bar{H} P_k^\top)}} \left\| \sqrt{\bar{H}}(x_k - \bar{x}) \right\| - \sqrt{1 - \frac{\lambda_{\min}(\bar{H})}{2\lambda_{\max}(P_k \bar{H} P_k^\top)}} \left\| \sqrt{\bar{H}}(x_k - \bar{x}) \right\| \geq o(\|x_k - \bar{x}\|),$$

we have that

$$\left\| \sqrt{\bar{H}}(x_{k+1} - \bar{x}) \right\| \leq \sqrt{1 - \frac{\lambda_{\min}(\bar{H})}{4\lambda_{\max}(P_k \bar{H} P_k^\top)}} \left\| \sqrt{\bar{H}}(x_k - \bar{x}) \right\|. \tag{D.6.6}$$

Notice that the condition above is implied when

$$\sqrt{1 - \frac{\lambda_{\min}(\bar{H})}{4\lambda_{\max}(P_k \bar{H} P_k^\top)}} - \sqrt{1 - \frac{\lambda_{\min}(\bar{H})}{2\lambda_{\max}(P_k \bar{H} P_k^\top)}} \geq \frac{o(\|x_k - \bar{x}\|)}{\sqrt{\lambda_{\min}(\bar{H})}\|x_k - \bar{x}\|}.$$

Thus, we obtain (D.6.6) for sufficiently large $k$.

Since we have, by Lemma D.10, $\lambda_{\max}(P_k \bar{H} P_k^\top) \leq \left(\sqrt{\frac{n}{s}} + \mathcal{C}\right)^2 \lambda_{\max}(\bar{H})$ with probability at least $1 - 2\exp(-s)$, the upper bound can be rewritten as

$$\left\| \sqrt{\bar{H}}(x_{k+1} - \bar{x}) \right\| \leq \sqrt{1 - \frac{\lambda_{\min}(\bar{H})}{4\lambda_{\max}(\bar{H})(\sqrt{n/s} + \mathcal{C})^2}} \left\| \sqrt{\bar{H}}(x_k - \bar{x}) \right\|.$$

Since the probabilistically valid properties that we used in this proof are Lemmas B.2, D.8, D.10 and D.9, the probability lower bound is given by $1 - 2\exp\left(-\frac{C}{4}s\right) - 8\exp(-s)$. $\quad\square$

### D.6.2 Local quadratic convergence

Let $\bar{y} = R\bar{x}$, where we recall that $\bar{x}$ be the strict local minimizer of $f$. Then the following properties hold:

$$f(x) = f(R^\top R x) = l(Rx),$$
$$\nabla f(x) = \nabla f(\Pi x) = \Pi^\top \nabla f(x) = \Pi \nabla f(x), \tag{D.6.7}$$
$$\|\nabla f(x)\| = \sqrt{\|\nabla f(x)^\top \nabla f(x)\|^2} = \sqrt{\|\nabla f(x)^\top \Pi \nabla f(x)\|} = \|R\nabla f(x)\|,$$
$$\exists \rho > 0 \text{ s.t. } \|\nabla l(y)\| \geq \rho\|y - \bar{y}\|,$$
$$\exists \gamma > 0 \text{ s.t. } \|\nabla f(x)\| \geq \gamma\|R(x - \bar{x})\| \quad (\gamma = \sigma_{\min}(R^\top)\rho). \tag{D.6.8}$$

Next, we show some lemmas regarding Lipschitz continuity.

**Lemma D.11.** *Suppose Assumptions 1 and 4 hold. There exist constants $L_l, M_l > 0$ such that for any $y_1, y_2 \in \mathbb{R}^r$, the following inequalities hold:*

$$\|\nabla l(y_1) - \nabla l(y_2)\| \leq L_l \|y_1 - y_2\|, \tag{D.6.9}$$
$$\|\nabla^2 l(y_1) - \nabla^2 l(y_2)\| \leq M_l \|y_1 - y_2\|. \tag{D.6.10}$$

*Proof.* Since $l(y) = f(R^\top y)$, by the Lipschitz continuity of $f$, we have

$$\|\nabla l(y_1) - \nabla l(y_2)\| = \left\| R\,\nabla f(x)|_{x=R^\top y_1} - R\,\nabla f(x)|_{x=R^\top y_2} \right\|$$
$$\leq \|R\| \left\| \nabla f(x)|_{x=R^\top y_1} - \nabla f(x)|_{x=R^\top y_2} \right\|$$
$$\leq \|R\| L \left\| R^\top y_1 - R^\top y_2 \right\|$$
$$\leq \|R\|^2 L \|y_1 - y_2\|.$$

Moreover, we have

$$\|\nabla^2 l(y_1) - \nabla^2 l(y_2)\| = \left\| R\left( \nabla^2 f(x)|_{x=R^\top y_1} - \nabla^2 f(x)|_{x=R^\top y_2} \right) R^\top \right\|$$
$$\leq \|R\|^2 \left\| \nabla^2 f(x)|_{x=R^\top y_1} - \nabla^2 f(x)|_{x=R^\top y_2} \right\|$$
$$\leq \|R\|^2 M \left\| R^\top y_1 - R^\top y_2 \right\|$$
$$\leq \|R\|^3 M \|y_1 - y_2\|.$$

Therefore, by setting $L_l = \|R\|^2 L$ and $M_l = \|R\|^3 M$, the lemma is proved. $\quad\square$

**Lemma D.12.** *Suppose Assumptions 1 and 4 hold. There exists a constant $L_\Pi > 0$ such that for any $x_1, x_2 \in \mathbb{R}^n$, the following inequality holds:*

$$\|\nabla f(x_1) - \nabla f(x_2)\|_\Pi \leq L_\Pi \|x_1 - x_2\|_\Pi.$$

*Proof.* Since

$$
\begin{aligned}
\|\nabla f(x_1) - \nabla f(x_2)\|_\Pi &= \sqrt{\left\|(\nabla f(x_1) - \nabla f(x_2))^\top \Pi \left(\nabla f(x_1) - \nabla f(x_2)\right)\right\|} \\
&= \sqrt{\left\|(\nabla f(x_1) - \nabla f(x_2))^\top \left(\Pi \nabla f(x_1) - \Pi \nabla f(x_2)\right)\right\|} \\
&= \sqrt{\left\|(\nabla f(x_1) - \nabla f(x_2))^\top \left(\nabla f(x_1) - \nabla f(x_2)\right)\right\|} \quad \text{(by (D.6.7))} \\
&\leq \|\nabla f(x_1) - \nabla f(x_2)\| \\
&= \left\|R^\top \left(\nabla l(y)|_{y=Rx_1} - \nabla l(y)|_{y=Rx_2}\right)\right\| \\
&\leq L_l \|R\| \|Rx_1 - Rx_2\|,
\end{aligned}
$$

the lemma is proved by setting $L_\Pi = L_l \|R\|$. $\qquad\square$

From this lemma, it immediately follows that $\|g_k\| = \|g_k\|_\Pi = O(\|x_k - \bar{x}\|_\Pi)$.

In the following, we analyze the convergence rate of $\|x_k - \bar{x}\|_\Pi$, which leads to the convergence rate of $f(x_k) - f(\bar{x})$. First, note that $t \neq 0$ *(a.s.)* is derived from the fact that the Hessian is positive semi-definite near $\bar{x}$. However, the proof is omitted since it is similar to that of Lemma D.8. This implies $d_k = P_k^\top \frac{\tilde{v}_k}{t_k}$.

Next, we show two auxiliary lemmas.

**Lemma D.13.** *Suppose Assumption 4 holds. Then,*

$$\|H_k d_k + g_k\| \leq \frac{\sqrt{r/s} + \mathcal{C}}{\zeta \left(1 - \sqrt{\frac{r-1}{s}}\right)} \|g_k\|_\Pi \|d_k\|_\Pi^2$$

*holds for $r = \mathrm{rank}(\Pi)$, a universal constants $\bar{C}$ and $\bar{c}$, and any $\zeta > 0$ with probability at least $1 - 2\exp(-r) - (\bar{C}\zeta)^{s-r+1} - e^{-\bar{c}s}$.*

*Proof.* Let $U_R$ be an orthogonal matrix whose first $r$ rows are given by $R$. It follows that $U_R P_k^\top$ has the same distribution as $P_k^\top$, meaning that each element of $U_R P_k^\top$ is distributed according to $\mathcal{N}(0, 1/s)$. As $RP_k^\top$ is the first $r$ rows of $U_R P_k^\top$, $RP_k^\top$ is an $r \times s$ random matrix with elements independently drawn from $\mathcal{N}(0, 1/s)$. Define $\tilde{P}_k^\top := RP_k^\top$, using (D.4.1), we have

$$
\begin{aligned}
\|H_k d_k + g_k\| &\leq \frac{1}{\sigma_{\min}(\tilde{P}_k)} \|P_k(H_k d_k + g_k)\| \\
&\leq \frac{\sigma_{\max}(\tilde{P}_k)}{\sigma_{\min}(\tilde{P}_k)} \|RP_k^\top P_k(H_k d_k + g_k)\|.
\end{aligned}
$$

From (Rudelson & Vershynin, 2009, Theorem 1.1), we have

$$\forall \zeta > 0, \quad \Pr\left[\sigma_{\min}\left(\tilde{P}_k\right) \geq \zeta \left(1 - \sqrt{\frac{r-1}{s}}\right)\right] \geq 1 - (\bar{C}\zeta)^{s-r+1} - e^{-\bar{c}s}$$

for some constants $\bar{C}, \bar{c}$ and from Lemma B.2, we have

$$\Pr\left[\sigma_{\max}(\tilde{P}_k) \leq \sqrt{\frac{s}{r}} + \mathcal{C}\right] \geq 1 - 2\exp(-r).$$

Therefore, for any $\zeta > 0$, the following inequality holds with probability at least $1 - 2\exp(-r) - (\bar{C}\zeta)^{s-r+1} - e^{-\bar{c}s}$.

$$\|H_k d_k + g_k\| \leq \frac{\sqrt{r/s} + C}{\zeta\left(1 - \sqrt{\frac{r-1}{s}}\right)} \|RP_k^\top P_k (H_k d_k + g_k)\|. \tag{D.6.11}$$

By multiplying $RP_k^\top$ to both sides of (B.2.8), we obtain

$$RP_k^\top P_k (H_k d_k + g_k) = -\theta_k R d_k. \tag{D.6.12}$$

Thus we have

$$\begin{aligned}
\|H_k d_k + g_k\| &\leq \frac{\sqrt{r/s} + C}{\zeta\left(1 - \sqrt{\frac{r-1}{s}}\right)} \|RP_k^\top P_k (H_k d_k + g_k)\| \quad \text{(by (D.6.11))} \\
&= \frac{\sqrt{r/s} + C}{\zeta\left(1 - \sqrt{\frac{r-1}{s}}\right)} \| - \theta_k R d_k\| \quad \text{(by (D.6.12))} \\
&= \frac{\sqrt{r/s} + C}{\zeta\left(1 - \sqrt{\frac{r-1}{s}}\right)} \theta_k \|R d_k\| \\
&= \frac{\sqrt{r/s} + C}{\zeta\left(1 - \sqrt{\frac{r-1}{s}}\right)} \theta_k \|d_k\|_\Pi.
\end{aligned} \tag{D.6.13}$$

Moreover, from (B.2.8) and $\delta = 0$, we have

$$\begin{aligned}
\theta_k &= -g_k^\top d_k \\
&= -(\Pi g_k)^\top d_k \\
&= -(R g_k)^\top (R d_k) \\
&\leq \|g_k\|_\Pi \|d_k\|_\Pi.
\end{aligned} \tag{D.6.14}$$

Combining (D.6.13) and (D.6.14), we obtain

$$\|H_k d_k + g_k\| \leq \frac{\sqrt{r/s} + C}{\zeta\left(1 - \sqrt{\frac{r-1}{s}}\right)} \|g_k\|_\Pi \|d_k\|_\Pi^2.$$

$\square$

**Lemma D.14.** *Suppose Assumptions 1 and 4 hold. The following inequality holds:*

$$\|g_{k+1} - (H_k d_k + g_k)\| \leq \frac{1}{2} M_l \|d_k\|_\Pi^2.$$

*Proof.* By considering the Taylor expansion of $t \mapsto \nabla f(x_k + td_k)$, we obtain the following equation:

$$g_{k+1} = g_k + \int_0^1 \nabla^2 f(x_k + td_k) d_k dt.$$

By subtracting $H_k d_k + g_k$ from both sides, we obtain

$$g_{k+1} - (H_k d_k + g_k) = \int_0^1 \left(\nabla^2 f(x_k + td_k) - H_k\right) d_k dt.$$

By evaluating the norm of both sides, we obtain the following inequality:

$$
\begin{aligned}
\|g_{k+1} - (H_k d_k + g_k)\| &\leq \int_0^1 \left\| \left(\nabla^2 f(x_k + t d_k) - H_k\right) d_k \right\| dt \\
&= \int_0^1 \left\| R^\top \left( \nabla^2 l(y)\big|_{y=R(x_k + t d_k)} - \nabla^2 l(y)\big|_{y=R x_k} \right) R d_k \right\| dt \\
&\leq \int_0^1 \|R\| \left\| \left( \nabla^2 l(y)\big|_{y=R(x_k + t d_k)} - \nabla^2 l(y)\big|_{y=R x_k} \right) \right\| \|R d_k\| dt \\
&\leq \|R\| \int_0^1 M_l \|R(x_k + t d_k) - R x_k\| \|R d_k\| dt \quad \text{(by (D.6.10))} \\
&= M_l \|R\| \|R d_k\|^2 \int_0^1 t dt \\
&= \frac{1}{2} M_l \|R\| \|d_k\|_\Pi^2 .
\end{aligned}
$$

Thus, the lemma is proved. $\qquad \square$

**Proof of Theorem 3.4**

*Proof.* At first, we will show the first inequality in Theorem 3.4. The following inequality holds with probability at least $1 - 2\exp(-r) - (\bar{C}\zeta)^{s-r+1} - e^{-\bar{c}s}$ for any $\zeta > 0$:

$$
\begin{aligned}
\|x_{k+1} - \bar{x}\|_\Pi &\leq \frac{1}{\gamma} \|g_{k+1}\| \quad \text{(by (D.6.8))} \\
&\leq \frac{1}{\gamma} \left( \|H_k d_k + g_k\| + \|g_{k+1} - (H_k d_k + g_k)\| \right) \\
&\leq \frac{1}{\gamma} \left( \frac{\sqrt{r/s} + \mathcal{C}}{\zeta \left(1 - \sqrt{\frac{r-1}{s}}\right)} \|g_k\|_\Pi \|d_k\|_\Pi^2 + \frac{1}{2} M_l \|R\| \|d_k\|_\Pi^2 \right) \quad \text{(by Lemma D.13 and Lemma D.14)} \\
&= \frac{1}{\gamma} \left( \frac{\sqrt{r/s} + \mathcal{C}}{\zeta \left(1 - \sqrt{\frac{r-1}{s}}\right)} \|x_k - \bar{x}\|_\Pi + \frac{1}{2} M_l \|R\| \right) \|d_k\|_\Pi^2 \quad \text{(by } \|g_k\|_\Pi = O(\|x_k - \bar{x}\|_\Pi)).
\end{aligned}
$$
(D.6.15)

Now, we have

$$
\begin{aligned}
\|d_k\|_\Pi &\leq \|x_{k+1} - \bar{x}\|_\Pi + \|x_k - \bar{x}\|_\Pi \\
&\leq \frac{1}{\gamma} \left( \frac{\sqrt{r/s} + \mathcal{C}}{\zeta \left(1 - \sqrt{\frac{r-1}{s}}\right)} \|x_k - \bar{x}\|_\Pi + \frac{1}{2} M_l \|R\| \right) \|d_k\|_\Pi^2 + \|x_k - \bar{x}\|_\Pi ,
\end{aligned}
$$

which can be rearranged as

$$
\left(1 - \frac{\|d_k\|_\Pi}{\gamma} \left( \frac{\sqrt{r/s} + \mathcal{C}}{\zeta \left(1 - \sqrt{\frac{r-1}{s}}\right)} \|x_k - \bar{x}\|_\Pi + \frac{1}{2} M_l \|R\| \right)\right) \|d_k\|_\Pi \leq \|x_k - \bar{x}\|_\Pi .
$$

Since $\|x_k - \bar{x}\|_\Pi \to 0$ and $\|d_k\|_\Pi \to 0$, for sufficiently large $k$, we have $\frac{1}{2}\|d_k\|_\Pi \leq \|x_k - \bar{x}\|_\Pi$. Combining this with (D.6.15), for sufficiently large $k$, we obtain

$$
\begin{aligned}
\|x_{k+1} - \bar{x}\|_\Pi &\leq \frac{4}{\gamma} \left( \frac{\sqrt{r/s} + \mathcal{C}}{\zeta \left(1 - \sqrt{\frac{r-1}{s}}\right)} \|x_k - \bar{x}\|_\Pi + \frac{1}{2} M_l \|R\| \right) \|x_k - \bar{x}\|_\Pi^2 \\
&\leq \frac{4 M_l \|R\|}{\gamma} \|x_k - \bar{x}\|_\Pi^2 \quad \text{(since } k \text{ is sufficiently large).}
\end{aligned}
$$

Therefore, we have derived the first statement in Theorem 3.4. Next, we move to the next inequality in Theorem 3.4.

$$
\begin{aligned}
f(x_k) - f(\bar{x}) &= \int_0^1 \left( \nabla f(\bar{x} + t(x_k - \bar{x})) - \nabla f(\bar{x}) \right)^\top (x_k - \bar{x}) dt \\
&= \int_0^1 \left( \nabla l(y)|_{y=R(\bar{x}+t(x_{k+1}-\bar{x}))} - \nabla l(y)|_{y=R\bar{x}} \right)^\top R(x_{k+1} - \bar{x}) dt.
\end{aligned}
$$
(D.6.16)

(D.6.16) is bounded above by

$$
\begin{aligned}
\int_0^1 \left\| \nabla l(y)|_{y=R(\bar{x}+t(x_k-\bar{x}))} - \nabla l(y)|_{y=R\bar{x}} \right\| \|R(x_k - \bar{x})\| \, dt \\
\leq L_l \|R(x_k - \bar{x})\|^2 \int_0^1 t \, dt \quad \text{(by (D.6.9))} \\
= \frac{L_l}{2} \|x_k - \bar{x}\|_\Pi^2.
\end{aligned}
$$
(D.6.17)

(D.6.16) is bounded below by

$$
\begin{aligned}
\left| \int_0^1 \left( \nabla l(y)|_{y=R(\bar{x}+t(x_k-\bar{x}))} - \nabla l(y)|_{y=R\bar{x}} \right)^\top \frac{tR(x_k - \bar{x})}{t} dt \right| \\
\geq \int_0^1 \frac{2\rho \|tR(x_k - \bar{x})\|^2}{t} dt \quad \text{(by strong convexity of } l\text{)} \\
= 2\rho \|x_k - \bar{x}\|_\Pi^2 \int_0^1 t \, dt \\
= \rho \|x_k - \bar{x}\|_\Pi^2.
\end{aligned}
$$
(D.6.18)

We are now ready to prove the theorem.

$$
\begin{aligned}
f(x_{k+1}) - f(\bar{x}) &\leq \frac{L_l}{2} \|x_{k+1} - \bar{x}\|_\Pi^2 \quad \text{(by (D.6.17))} \\
&\leq \frac{8 L_l M_l^2 \|R\|^2}{\gamma^2} \|x_k - \bar{x}\|_\Pi^4 \quad \text{(by Theorem 3.4)} \\
&\leq \frac{8 L_l M_l^2 \|R\|^2}{\gamma^2 \rho^2} (f(x_k) - f(\bar{x}))^2. \quad \text{(by (D.6.18))}
\end{aligned}
$$

We recall that all of the above hold with probability at least $1 - 2\exp(-r) - (\bar{C}\zeta)^{s-r+1} - e^{-\bar{c}s}$ for any $\zeta > 0$. If we set $\zeta$ sufficiently small, the probability is bounded from below by $1 - 3\exp(-r) - e^{-\bar{c}s}$, which ends the proof. $\qquad\square$

## E    Convergence theorems under Algorithm 6

We provide additional theoretical considerations regarding subspace dimension $s$ such as $s < \Omega(\log n)$. In practice, $s = \Omega(\log n)$ is sufficiently small, and Algorithm 1 works effectively; this analysis is primarily of theoretical interest.

While setting $s = o(\log n)$ means that the success probability of each iteration is no longer high, the algorithm can simply retry until success. Specifically, in this section, we present a slight modification (see Algorithm 6) of Algorithm 1, where line 11-line 15 are added to decrease the value of the objective function $f$ at every iteration. For Algorithm 1, we can prove the convergence to an $\varepsilon$-FOSP with arbitrarily high probability under the same hypothesis even for small $s$ independent of the dimension $n$ ($s$ needs only to be greater than some constant). Notice that all the results proved for Algorithm 1 also hold for Algorithm 6. This is because the probability that the function decreases at each iteration is already taken into account in the probabilistic results that we prove.

---

**Algorithm 6** RSHTR: Random Subspace Homogenized Trust Region Method (variant)

---
1: **function** RSHTR$(s, n, \delta, \Delta, \max\_iter)$
2:     global_mode $=$ True
3:     **for** $k = 1, \ldots, \max\_iter$ **do**
4:         $P_k \leftarrow s \times n$ random Gaussian matrix with each element being from $\mathcal{N}(0, 1/s)$
5:         $\tilde{g}_k \leftarrow P_k g_k$
6:         $(t_k, \tilde{v}_k) \leftarrow$ optimal solution of (2.2.1) by eigenvalue computation
7:         $d_k \leftarrow \begin{cases} P_k^\top \tilde{v}_k / t_k, & \text{if } t_k \neq 0 \\ P_k^\top \tilde{v}_k, & \text{otherwise} \end{cases}$
8:         **if** global_mode and $\|d_k\| > \Delta$ **then**
9:             $\eta_k \leftarrow \Delta / \|d_k\|$                  ▷ or get from backtracking line search
10:             $y_{k+1} \leftarrow x_k + \eta_k d_k$
11:             **if** $f(y_{k+1}) < f(x_k)$ **then**
12:                 $x_{k+1} = y_{k+1}$
13:             **else**
14:                 $x_{k+1} = x_k$
15:             **end if**
16:         **else**
17:             $x_{k+1} \leftarrow x_k + d_k$
18:             **terminate**        ▷ or continue with $(\delta, \text{global\_mode}) \leftarrow (0, \text{False})$ for local
    convergence

---

**Theorem E.1** (Global convergence to an $\varepsilon$–FOSP). *Suppose that Assumption 1 holds. Let*

$$0 < \varepsilon \leq \tfrac{M^2}{8}, \quad \delta = \left(\sqrt{\tfrac{n}{s}} + \mathcal{C}\right)^2 \sqrt{\varepsilon} \quad and \quad \Delta = \tfrac{\sqrt{\varepsilon}}{M}.$$

*Then modified RSHTR (Algorithm 6) outputs an $\varepsilon$–FOSP in at most $O\left(\varepsilon^{-3/2}\right)$ iterations with probability at least*

$$1 - \exp\left(-\frac{1}{8}(1 - \delta_s)U_\varepsilon\right) - 4\exp\left(-\frac{C}{4}s\right) - 4\exp\left(-s\right),$$

*where $\mathcal{C}$ and $C$ are absolute constants, $U_\varepsilon := \lfloor \frac{6}{\delta_s} M^2 \left(f(x_0) - \inf_{x \in \mathbb{R}^n} f(x)\right) \varepsilon^{-3/2} \rfloor + 1$, and $\delta_s := 1 - 2\exp(-s)$.*

*Proof.* Let us consider how many times we iterate in the case where $|d_k| > \Delta$ at most. According to Lemma D.1 and Lemma D.2, the objective function decreases by at least

$$\frac{1}{2\left(\sqrt{n/s} + \mathcal{C}\right)^2}\Delta^2\delta - \frac{M}{6}\Delta^3 = \frac{\varepsilon^{3/2}}{3M^2}$$

with probability at least $1 - 2\exp\left(-s\right)$. Let $Y_k \in \{0, 1\}$ be a random variable equal to 1 if and only the objective function decreases at least by the above quantity. Then, after $K$ iterations, the objective function decreases by at least:

$$\frac{\varepsilon^{3/2}}{3M^2} \sum_{k=1}^{K} Y_k.$$

Since, for all $k$, $\mathbb{E}[Y_k] \geq 1 - 2\exp(-s) := 1 - \delta_s$, we have by a Chernoff bound (see Vershynin (2018)) that for all $\delta \in (0, 1)$,

$$\mathbb{P}\left(\sum_{k=1}^{K} Y_k \geq (1 - \delta)(1 - \delta_s)K\right) \geq 1 - \exp\left(-\frac{\delta^2}{2}(1 - \delta_s)K\right).$$

Hence with probability at least $1 - \exp\left(-\frac{1}{8}(1 - \delta_s)K\right)$, after $K$ iterations, the objective function decreases by at least

$$\frac{\varepsilon^{3/2}}{6M^2}(1 - 2\exp(-s))K.$$

Since the total amount of decrease does not exceed $D := f(x_0) - \inf_{x \in \mathbb{R}^n} f(x)$, we deduce that the number of iterations for the case where $\|d_k\| > \Delta$ is at most $\lfloor \frac{6M^2 D \varepsilon^{-3/2}}{1 - 2\exp(-s)} \rfloor$. Also, since the algorithm terminates once it enters the case $\|d_k\| \le \Delta$, the total number of iterations is at most $U_\varepsilon = \lfloor \frac{6M^2 D \varepsilon^{-3/2}}{1 - 2\exp(-s)} \rfloor + 1$ at least.

We can compute an $\varepsilon$–FOSP with probability at least $1 - 4\exp\left(-\frac{C}{4}s\right) - 4\exp(-s)$, which can be easily checked by applying Lemma 3.2 to the given $\delta$ and $\Delta$.

Therefore, RSHTR converges in $\lfloor \frac{6M^2 D \varepsilon^{-3/2}}{1 - 2\exp(-s)} \rfloor + 1 = O(\varepsilon^{-3/2})$ iterations with probability at least

$$1 - \exp\left(-\frac{1}{8}(1 - \delta_s)U_\varepsilon\right) - 4\exp\left(-\frac{C}{4}s\right) - 4\exp(-s),$$

where $U_\varepsilon = \lfloor \frac{6M^2 D \varepsilon^{-3/2}}{1 - 2\exp(-s)} \rfloor + 1$. $\qquad\square$

## F    Experimental Details

Throughout all experiments, the parameters of the algorithms were set as follows:

- HSODM: $(\delta, \Delta, \nu) = (10^{-3}, 10^{-3}, 10^{-1})$
- RSGD: $s = 100$
- RSRN: $(\gamma, c_1, c_2, s) = (1/2, 2, 1, 100)$
- RSHTR: $(\delta, \Delta, \nu, s) = (10^{-3}, 10^{-3}, 10^{-1}, 100)$

Here, we denote the dimensionality of subspace as $s$.

The datasets and other details of each task are described below.

### F.1    Matrix factorization

In this task, no preprocessing is performed. We chose 50 as the feature dimension $k$. Here is the dataset we used for this task.

**MovieLens 100k (Harper & Konstan, 2015)**

- Shape of $R$: (943, 1682)
- Problem dimension: 131,250
- Source: downloaded using scikit-learn (Pedregosa et al., 2011)

### F.2    Logistic regression

In this task, all datasets were preprocessed as follows:

- 10,000 features were selected to limit the problem dimensionality for the datasets with features more than 10,000.
- 1,000 samples were selected to save the computational resource for the datasets with samples of more than 1,000.

Here is a list of the datasets we used for this task.

**news20.binary (Kogan et al., 2009)**

- Problem dimension: 10,001
- Source: `http://www.csie.ntu.edu.tw/~cjlin/libsvmtools/datasets/binary.html`

**rcv1.binary (Lewis et al., 2004)**

- Problem dimension: 10,001
- Source: `http://www.csie.ntu.edu.tw/~cjlin/libsvmtools/datasets/binary.html`

**Internet Advertisements (Kushmerick, 1998)**

- Problem dimension: 1,558
- Source: `https://archive.ics.uci.edu/dataset/51/internet+advertisements`

### F.3 SOFTMAX REGRESSION

In this task, all datasets were preprocessed in the same way as F.2.

- For datasets with more than 10,000 features, the first 10,000 features were selected to limit problem dimensionality.
- For datasets with more than 1,000 samples, 1,000 samples were selected to conserve computational resources.

Here is a list of the datasets we used for this task.

**news20 (Lang, 1995)**

- Number of classes: 20
- Problem dimension: 200,020
- Source: `https://www.csie.ntu.edu.tw/~cjlin/libsvmtools/datasets/multiclass.html`

**SCOTUS (Chalkidis et al., 2021)**

- Number of classes: 13
- Problem dimension: 130,013
- Source: `https://www.csie.ntu.edu.tw/~cjlin/libsvmtools/datasets/multiclass.html`

### F.4 DEEP NEURAL NETWORKS

In this task, we used a 16-layer fully connected neural network with bias terms and the widths of each layer are:

$$[\texttt{input\_dim}, 128, 64, 32, 32, 32, 32, 32, 32, 32, 32, 32, 32, 32, 32, \texttt{output\_dim}]$$

We utilized subsets of 1,000 images from each of the following datasets. Here is a list of the datasets we used for this task. When using the test data, we sampled an additional 1000 data points.

**MNIST (Deng, 2012)**

- Input dimension: $28 \times 28 = 784$
- Output dimension: 10
- Problem dimension: 123,818
- Source: downloaded using scikit-learn (Pedregosa et al., 2011)

**CIFAR-10 (Krizhevsky, 2009)**

- Input dimension: $32 \times 32 \times 3 = 3,076$
- Output dimension: 10
- Problem dimension: 416,682
- Source: downloaded using scikit-learn (Pedregosa et al., 2011)

## G ADDITIONAL NUMERICAL EXPERIMENTS

In all experiments in this section, the proposed method exhibited the fastest convergence.

**Matrix factorization with mask (MFM):** We first recall matrix factorization (without mask) formulation:

$$\min_{U \in \mathbb{R}^{n_u \times k}, V \in \mathbb{R}^{k \times n_v}} \|UV - R\|_F^2 / (n_u n_v),$$

where $R \in \mathbb{R}^{n_u \times n_v}$ and $\|\cdot\|_F^2$ denotes the squared Frobenius norm. The masked version of the problem introduces a mask matrix $X \in \{0, 1\}^{n_u \times n_v}$ to handle missing entries in $R$. Using this mask matrix, MFM is formulated as

$$\min_{U \in \mathbb{R}^{n_u \times k}, V \in \mathbb{R}^{k \times n_v}} \|(UV - R) \odot X\|_F^2 / (n_u n_v),$$

where $X_{ij} = 1$ if $R_{ij}$ is not null and $X_{ij} = 0$ otherwise and the symbol $\odot$ denotes element-wise multiplication. The problem dimension is calculated as $n = (n_u + n_v)k$. The result of MFM on MovieLens 100k dataset (Harper & Konstan, 2015) is shown in Figure 5a.

**Classification:** The formulation of this task is the same as Section 4. The result of logistic regression on the Internet Advertisement dataset (Kushmerick, 1998) is shown in Figure 5b.

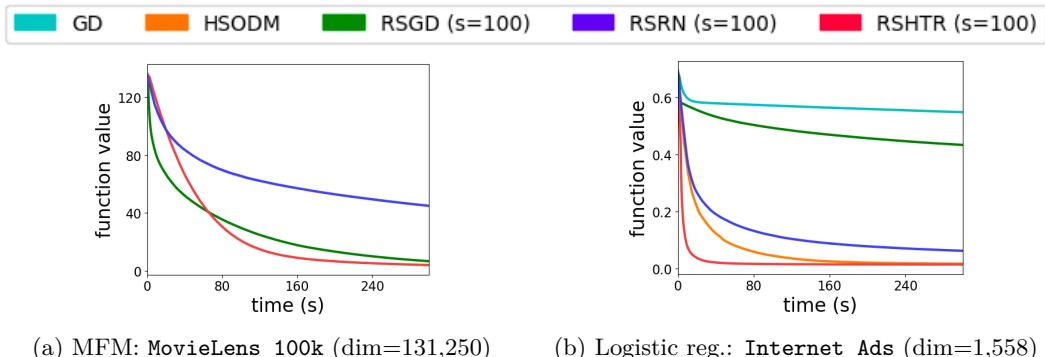

(a) MFM: `MovieLens 100k` (dim=131,250)     (b) Logistic reg.: `Internet Ads` (dim=1,558)

Figure 5: Comparison of our method to existing methods regarding the function value v.s. computation time. Each plot shows the average $\pm$ the standard deviation for five runs. Algorithms that did not complete a single iteration within the time limit are omitted.

**Classification Accuracy** To compare several methods from perspectives other than the loss function, we also evaluated train and test classification accuracies at the end of training (4,000 seconds for MNIST and 16,000 seconds for CIFAR-10), where training was stopped due to the time limit. The results are presented in Table 2. While the limited training data restricts generalization and results in moderate test accuracy, our approach still demonstrates superior performance compared to other methods.

Table 2: Train and test accuracies on MNIST and CIFAR-10 datasets. Our method, RSHTR, achieved the highest accuracy on both datasets.

| | MNIST | | CIFAR-10 | |
|---|---|---|---|---|
| Algorithm | Train | Test | Train | Test |
| RSGD | 0.412 | 0.355 | 0.315 | 0.146 |
| RSRN | 0.735 | 0.376 | 0.523 | 0.150 |
| **RSHTR** | **0.998** | **0.680** | **0.959** | **0.211** |

**Comparison with Heuristic Algorithms** We conducted further experiments to evaluate the performance of our algorithm against popular heuristics used in training deep neural networks. Specifically, we compared our proposed method with Adam (Kingma, 2014) and AdaGrad (Duchi et al., 2011) on the MNIST and CIFAR-10 datasets for classification using a neural network. The formulation of the task is the same as Section 4. What differs is that,

to ensure a meaningful comparison with fast optimization methods beyond random subspace methods, we adjusted the subspace dimension of our proposed algorithm to a smaller value than used in Section 4. The result and the hyperparameter settings are shown in Figure 6. While our proposed method does not surpass Adam in terms of convergence speed, it demonstrates superior numerical stability. This is likely attributed to the method's ability to avoid directions with rapidly increasing gradient norm by utilizing Hessian information. Furthermore, our method can match or outperform AdaGrad in convergence speed depending on the parameter settings. Specifically, when using numerically stable parameters for Ada-Grad, our method exhibits a faster convergence rate. Moreover, our method saves time to tune hyperparameters due to its consistent stability across different hyperparameter choices. Conversely, Adam and AdaGrad can become drastically unstable with increased learning rates aimed at faster convergence, necessitating trial and error for parameter optimization.

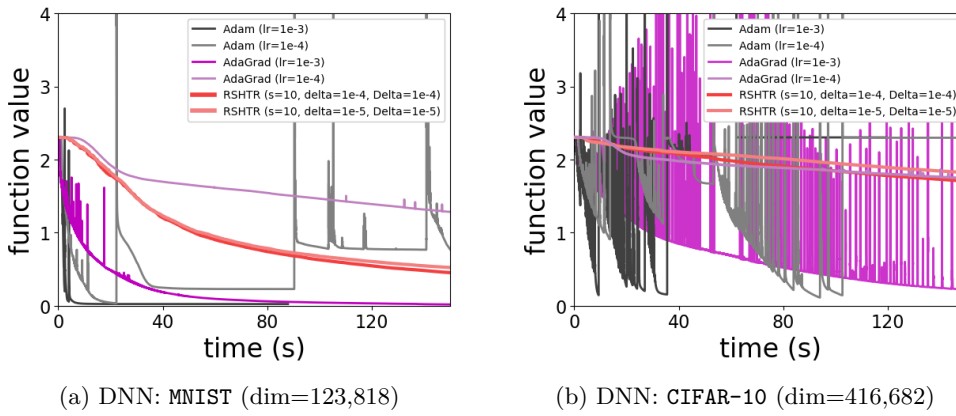

(a) DNN: `MNIST` (dim=123,818)  (b) DNN: `CIFAR-10` (dim=416,682)

Figure 6: Comparison of our method to heuristic algorithms.

