# OpenReview forum: "Improving Convergence Guarantees of Random Subspace Second-order Algorithm for Nonconvex Optimization"
_ICLR.cc/2025/Conference — ICLR 2025 Spotlight_

### Official Review · Reviewer_qDh3 · 2024-10-31

**Soundness:** 3
**Presentation:** 3
**Contribution:** 3
**Rating:** 8
**Confidence:** 3

**Summary:**

This paper offers an extension of current random subspace methods to  the Homogenized Trust Region method of Zhang et al. At each iteration, a random Gaussian matrix is sampled, and they use it to project the gradient and hessian. A trust region problem is then solved in the reduced space by an eigendecomposition. The theoretical analysis proves convergence to an $\epsilon$-FOSP in complexity $O(\epsilon^{-3/2})$, under the assumption of a Lipschitz gradient and Lipschitz Hessian. Under the further assumption of a low-rank Hessian at the output point, the authors prove convergence to an $\epsilon$-SOSP in complexity $O(\epsilon^{-3/2})$. Finally, under strict convexity, the authors prove local linear convergence, and under the assumption of a low-dimensional Hessian, the authors get quadratic convergence.

**Strengths:**

The main strengths of this work are reduced complexity, both in memory and computational time.  The high-probability bounds developed also point to a clear choice of dimension in $s\asymp \log n$. This is to make the probabilities vanishingly small in the dimension. The reduced trust region problem reduces to an eigenvalue problem, just as in past work by Zhang et al 2022, which can be solved efficiently in the lower dimension using standard methods. Overall, this represents an important addition to the nonconvex optimization literature.

**Weaknesses:**

There are a few weaknesses to this paper. First, the paper should be made more clear. For example, in theorems 3.3 and 3.4, it is unclear what “sufficiently large $k$” means. If the authors could provide statements that make this more precise, that would make the statements of the theorems stronger.

Some things are introduced without proper motivation or definition. For example, what problem does 2.2.1 solve? I know that it somehow comes from the past work and solves a trust region problem, but the paper would be improved with further discussion to motivate the problem.

In the section discussing overall complexity, the statements include an epsilon that references the later theorem on convergence to an $\epsilon$-FOSP. However, this theorem only applies under certain assumptions -- the authors should make clear in this section that they are examining complexity under these same assumptions.

Another weakness is that the theoretical innovations are not entirely clear. The authors should comment on the theoretical innovations within the text since proofs are pushed to the appendix. Is the theory just a mashup of Zhang et al. 2022 and the JL lemma, or is something more intricate involved?

The assumption of low-rankness in the SOSP and quadratic convergence results seem quite restrictive. While it may be necessary for the quadratic convergence, is Assumption 2 necessary, or is there some weaker condition that could lead to quadratic convergence?  A discussion of the proof techniques would help here.

**Questions:**

- Choosing $s= \Omega(n)$ seems to be the recommended dimensional choice in practice. If one ignores the high-probability bounds, could one show a convergence guarantee in expectation for smaller subspace dimensions under the various assumptions?
- I am a little confused about the probability in the theorem statements of linear and quadratic convergence. Is it that you get a linear/quadratic decrease for a certain iteration with this probability, or is it that this holds uniformly over iterations with this probability? I suspect it's the former, but I want it to be clear.

---

> ### Author Response · Authors · 2024-11-22
>
> **(W1) Explain what “sufficiently large $ k $” means and be more precise.**
>
> **Response:**
> By "sufficiently large $ k $," we mean that the iterate $ x_k $ belongs to a sufficiently small neighborhood, $ U $, of $ \bar{x} $. This is typically the context in which we discuss local convergence.
>
> Specifically, in the proof of Theorem 3.3 (Section D.5.1), we arrive at the following inequality:
> $$
> \left\| \sqrt{\bar{H}}(x_{k+1} - \bar{x}) \right\| \leq \sqrt{1 - \frac{\lambda_{\min}(\bar{H})}{2 \lambda_{\max}(P_k \bar{H} P_k^{\top})}} \left\| \sqrt{\bar{H}}(x_k - \bar{x}) \right\| + o(\|x_k - \bar{x}\|).
> $$
>
> Then, we take the neighborhood $ U $ sufficiently small such that:
> $$
> \sqrt{1 - \frac{\lambda_{\min}(\bar{H})}{2 \lambda_{\max}(P_k \bar{H} P_k^{\top})}} \left\| \sqrt{\bar{H}}(x_k - \bar{x}) \right\| + o(\|x_k - \bar{x}\|) \leq \sqrt{1 - \frac{\lambda_{\min}(\bar{H})}{4 \lambda_{\max}(P_k \bar{H} P_k^{\top})}} \left\| \sqrt{\bar{H}}(x_k - \bar{x}) \right\|.
> $$
> We added a simple explanation before Theorem 3.3 to clarify the meaning of $k$ large enough in addition to more arguments to the proof in line 1741-1760 so that one can see why sufficiently large $k$ is necessary.
> We also added a similar argument to the proof of Theorem 3.4.
>
>
> **(W2) Explain and motivate what problem 2.2.1 solves.**
>
> **Response:**
> Thank you for pointing out this omission. We have added further explanation in line 770 in Appendix A to clarify the motivation behind homogenization.
> We have also noted this addition below equation (2.2.1).
>
> **(W3) In the section discussing overall complexity, include the assumption needed about convergence to an $ \varepsilon $-FOSP.**
>
> **Response:**
> We have clarified this part by adding the following sentence in page 5 to the section:
> "Under gradient Lipschitz and Hessian Lipschitz assumptions (see Assumption 1)."
>
> **(W4) Explain better the theoretical innovation of the paper. Is it just a mashup of Zhang et al. (2022) and the JL lemma, or is something more intricate involved?**
>
> **Response:**
> Thank you for your constructive feedback. At first glance, this study might appear to be a simple combination of the random subspace method and HSODM. However, the main contribution of this work lies in the convergence analysis of the algorithm resulting from this combination, which involves nontrivial and novel analyses.
>
> First, regarding global convergence, simply combining the JL lemma (which addresses the preservation of norm when projecting from high-dimensional to low-dimensional spaces) with HSODM is insufficient to capture the iterative behavior of the algorithm and ensure convergence guarantees.  Global convergence requires  a more carefull analysis.
>
> Additionally, to ensure convergence to a second-order stationary point (SOSP), we have introduced a novel and essential lemma (Lemma 3.4) that establishes the relationship between the smallest eigenvalues of the full Hessian and the restricted Hessian. We believe this result is both nontrivial and interesting.
>
> Regarding local convergence, directly applying existing analytical techniques is challenging. As mentioned in the introduction, guaranteeing the local convergence rate is tricky. While full-space algorithms ensure quadratic convergence, our proposed algorithm exhibits linear convergence. This necessitates a fundamental change in the analytical approach, requiring a methodology distinct from HSODM.
>
> Furthermore, guaranteeing local superlinear convergence for low-effective functions is a completely new result not considered in HSODM. The analysis of how this class of functions behaves under the random subspace method represents an independent and novel contribution.
>
> Thus, while this paper may appear to be a straightforward combination of an existing method and an existing lemma, it involves multiple novel and intriguing contributions in its analysis. Stochastic GD serves as a similar example, showing that even a combination of existing methods can lead to new theoretical insights through in-depth analysis.
>
> Once again, thank you for your feedback.
>
>
> **(W5) Is the assumption of low-rankness in the SOSP and quadratic convergence too restrictive?**
>
> **Response:**
> No, the assumption of low-rankness is not too restrictive and includes many cases commonly observed in recent machine learning optimization problems.
> We have added an explanation below Assumption 2 to clarify that many problems actually satisfy it.

---

> > ### Author Response · Authors · 2024-11-22
> >
> > **(Q1) Choosing $ s = \Omega(n) $ seems to be the recommended dimensional choice in practice. Can you extend your results to smaller subspace dimensions in expectation?**
> >
> > **Response:**
> > Actually, in the current setting, $ s = O(\log(n)) $ is the recommended choice. We think it should be possible to prove that the global convergence result holds in expectation for any value of $s$, although we did not proved it yet. However notice that with a slight modification to the current algorithm (i.e., Algorithm 1), where we only accept iterations that decrease the function value (see Algorithm 5 in Section E), we could easily prove in Theorem E.1 that even for small $ s $ (where $ s $ needs only to be greater than a constant), Algorithm 5 can converge to an $ \varepsilon $-FOSP with arbitrarily high probability under the same assumptions.
> > We would like to emphasize that in all numerical computations performed, the sequence of iterates computed with RSHTR (Algorithm 1) was already non-increasing.
> >
> > **(Q2) I am a little confused about the probability in the theorem statements of linear and quadratic convergence. Is it that you get a linear/quadratic decrease for a certain iteration with this probability, or is it that this holds uniformly over iterations with this probability? I suspect it's the former, but I want it to be clear.**
> >
> > **Response:**
> > We have a linear/quadratic decrease for a certain iteration with a given probability. We have rephrased Theorems 3.3 and 3.4 to avoid ambiguity.
> > We appreciate your sharp point. This helped us improve the paper.

---

> ### Comment · Reviewer_qDh3 · 2024-11-26
>
> Thank you for your responses to my questions. I am increasing my score and recommend acceptance.

---

### Official Review · Reviewer_hJM7 · 2024-11-01

**Soundness:** 3
**Presentation:** 3
**Contribution:** 3
**Rating:** 8
**Confidence:** 2

**Summary:**

This paper studies algorithms for solving unconstrained non-convex optimization problems of the form $\min_{x\in \mathbb{R}^n}f(x)$ for a non-convex function $f:\mathbb{R}^n\rightarrow \mathbb{R}$, in particular algorithms for finding an $\epsilon$-approximate first order stationary point. The paper considers algorithms that first pick a random small dimensional subspace of $n$-dimensional space and search for the best update direction in that subspace. The main idea of the paper is to develop a random subspace trust region method, and the paper is the first to give a complexity analysis for this class of methods. They show that their algorithm converges to an $\epsilon$-approximate first order stationary point with a global convergence rate of $\epsilon^{-3/2}$ iterations as well as local quadratic convergence under the assumption that $f$ is strongly convex. The main idea behind their work to make the analysis of random subspace trust region methods feasible is the minimum eigenvalue formulation based analysis of the trust region subproblem previous work of Zhang et al (this is a trust region based algorithm without random projections). This approach makes it feasible to analyze the effect of random projections in trust region based methods.

**Strengths:**

The main strength of the paper is present the first theoretical analysis of random subspace based trust region methods for non-convex optimization problems. The guarantees also match the best known guarantees for random projection based methods for non-convex optimization.

**Weaknesses:**

One minor weakness of the paper is that in the experiments section, the comparison of the proposed algorithm is not done with popular heuristic optimization algorithms such as adagrad or adam for experiments on deep neural networks.

**Questions:**

I would be interested in knowing the performance of the proposed algorithms with popular heuristics for training deep neural networks such as adagrad or adam.

---

> ### Author Response · Authors · 2024-11-22
>
> **(W1+Q1) No comparison with popular heuristic optimization algorithms such as adagrad or adam for experiments on deep neural networks. Can this be compared?**
>
> **Response:**
> We appreciate your valuable sugguestion. The answer is Yes, and we have included additional comparative experiments comparing our algorithm's performance against Adam and AdaGrad on the MNIST and CIFAR-10 datasets in "Comparison with Heuristic Algorithms" in Appendix G. Results in Figure 6 show our method's superior numerical stability and competitive convergence speed against AdaGrad, while requiring less hyperparameter tuning.

---

> > ### Comment · Reviewer_hJM7 · 2024-11-24
> > **Thanks for the response.**
> >
> > Thanks a lot for the additional experiments, your time and effort is appreciated.

---

### Official Review · Reviewer_zCBT · 2024-11-02

**Soundness:** 3
**Presentation:** 2
**Contribution:** 3
**Rating:** 6
**Confidence:** 2

**Summary:**

This paper is primarily inspired by the HSODM method developed by Zhang et al. (2022) and extends their ideas to the domain of random subspace methods. The authors  propose a new approach called the Random Subspace Homogenized Trust Region (RSHTR) method. They establish global convergence guarantees under specific conditions and provide theoretical proofs for convergence to a second-order stationary point. Additionally, the paper demonstrates local linear and superlinear convergence under certain assumptions. Experiments conducted on real-world datasets confirm that RSHTR offers significant computational advantages and improved convergence properties over previous algorithms.

**Reference**: Chuwen Zhang, Dongdong Ge, Chang He, Bo Jiang, Yuntian Jiang, Chenyu Xue, and Yinyu Ye. A homogenous second-order descent method for nonconvex optimization. arXiv preprint arXiv:2211.08212, 2022.

**Strengths:**

- The paper is well-structured and provides a clear presentation of the proposed algorithm. It establishes a strong theoretical foundation, presenting the first theoretical results for random subspace methods that demonstrate quadratic convergence properties for certain classes of functions.
- The experimental setup is appropriate, with sufficient comparisons made against existing methods. The results clearly illustrate both the practical convergence and favorable computational complexity of the proposed algorithm.

**Weaknesses:**

- Despite the logical layout, the paper contains numerous typos and punctuation errors, which negatively affect readability and the overall presentation.
- The paper mentions several constants (e.g., $C, c,\bar{C}, \bar{c}$) but does not provide detailed information regarding them throughout the text and appendix. It would be beneficial to clarify what parameters these constants are associated with, as they are likely not arbitrary values.
- While the theoretical contributions are substantial, the authors note that it is also possible to provide theoretical guarantees by adopting a standard line search strategy in RSHTR to compute $\eta_k$.
- To enhance the practical significance of the theoretical guarantees, it would be valuable to include additional experiments, such as classification accuracy tasks, to further validate the algorithm's effectiveness. Nonetheless, the current experimental results are already quite strong. Additionally, Figure 5 in Appendix F is missing a title and legend.

**Questions:**

1. In line 43 and line 168, the notation "subspace $P_{k}^{\top}\mathbb{R}^{s}$" appears. Could the authors please confirm whether this should be corrected to "subspace $P_{k}\mathbb{R}^{n}$" and ensure that the correct notation is used consistently throughout the paper?
2. In Table 1,  could the authors provide a legend or explanation for the symbols "1" and "2" in the Local column? This would help clarify the meaning for readers and improve understanding of the information presented.
3. In line 170, the normal distribution $N(0,1/s)$ is used, with variance chosen as $1/s$? Could the authors provide additional context or a reference to explain this choice of variance, as it would assist readers who may be less familiar with random subspace methods?
4. For equation (2.2.1), the reference to the "homogenization trick" could be expanded. Could the authors either provide a citation or briefly explain how this trick is derived such as Zhang et al. (2022)?
5. In line 184, the authors state that "unlike HSODM, RSHTR allows for other eigensolvers beyond the Lanczos method." However, the experimental section does not specify which algorithm was used for solving. Could the authors clarify this point?
6. In line 255, I understand that HSODM requires calculating $n+1$ Hessian vector products, resulting in a computational complexity of $O(n^2)$, while RSHTR requires $s + 1$ Hessian vector products. However, if RSHTR does not need to store the Hessian, why does HSODM require storing it?
7. In lines 318 and 360, could the authors clarify the relationship between the gradient norm and the relative function values? This would help in understanding why the complexities are expressed as $(n/s)^{3/4}$ and $(n/s)^{3}$.
8. In Algorithm 2, I noticed a difference in the stopping criterion compared to HSODM in Zhang et al. (2022). Could the authors elaborate on the equivalence of the stopping criteria used in both algorithms?
9. In line 1480 and below, the notation $\sqrt{\bar{H}}$ is not defined. Could the authors provide clarification on this notation?

**Reference**: Chuwen Zhang, Dongdong Ge, Chang He, Bo Jiang, Yuntian Jiang, Chenyu Xue, and Yinyu Ye. A homogenous second-order descent method for nonconvex optimization. arXiv preprint arXiv:2211.08212, 2022.

The article contains several typographical and punctuation errors that require careful review by the authors. For example:

- Line 28: "as follows. " should be "as follows:".
- Line 110: "Local" should be "local".
- Lemma 3.4, lines 728, 736, 741, Lemma B.3, line 1345, and so on: unctuation errors are present. I recommend that the authors conduct a thorough review of the entire paper.
- Line 372 and 417:  "global_mode" should be "False".
- Line 1525: $\||A_3+A_4\||$ should be $\||A_3-A_4\||$.

---

> ### Author Response · Authors · 2024-11-22
>
> **(W1) Despite the logical layout, the paper contains numerous typos and punctuation errors, which negatively affect readability and the overall presentation.**
>
> **Response:**
> Thank you for pointing out the typos. We have carefully reviewed and corrected them to improve the readability and presentation of the paper. We did our best to reduce the typos as much as possible, and we also used several tools to check our paper, such as Grammarly.
>
> **(W2) Details concerning the constants**
>
> **Response:**
> It is important to note that, apart from $c$, all constants (i.e., $C, \mathcal{C}, \bar{C}, \bar{c}$) are absolute and do not depend on any parameters. Regarding $c$, it is related to rewriting $\exp(-s)$ as $n^{-c}$ when $s = c \log(n)$. It is a user-defined parameter, and from a theoretical perspective, taking $s = O(\log(n))$ is considered a good choice.
>
> Moreover, evaluating $C, \mathcal{C}, \bar{C}, \bar{c}$ can be possible with some effort. For example, in:
> *Wainwright MJ. Basic tail and concentration bounds. In: High-Dimensional Statistics: A Non-Asymptotic Viewpoint. Cambridge Series in Statistical and Probabilistic Mathematics. Cambridge University Press; 2019:21-57.*,
> one can compute $C = \frac{1}{8}$. We added some remarks (see footnotes in page 6 and page 7) in the paper for the curious reader that would like to compute these constants, and we stressed that they were absolute constants. Thank you for the remark.
>
> **(W3) Provide theoretical guarantees by adopting a standard line search strategy.**
>
> **Response:**
> Thank you for your comment. We have addressed this by adding a new subsection, titled "Analysis considering a line search strategy" in Appendix C. This section generalizes our results to include the use of a line search for step size determination.
>
> **(W4) Perform additional experiments on classification accuracy tasks, and missing title and legend on Figure 5.**
>
> **Response:**
> Thank you for pointing out the missing title and legend for Figure 5. We have fixed them.
> We have also included additional experiments for classification accuracy tasks (see Table 2 in Appendix G). The results suggest that our method outperforms existing random subspace methods in these tasks.

---

> > ### Author Response · Authors · 2024-11-22
> >
> > **(Q1) Confirm the meaning of $P_k^\top \mathbb{R}^s$ and that it is not misused in the paper.**
> >
> > **Response:**
> > $P_k^\top \mathbb{R}^s = \{x \in \mathbb{R}^n \mid x = P_k^\top u, \ u \in \mathbb{R}^s\}$ is the correct notation to denote an $s$-dimensional subspace of $\mathbb{R}^n$. On the other hand, $P_k \mathbb{R}^n$ would refer to a space equal to $\mathbb{R}^s$, and thus not represent an $s$-dimensional subspace of $\mathbb{R}^n$.
> >
> >
> >
> >
> > **(Q2) Clarify symbols in the local column of Table 1.**
> >
> > **Response:**
> > We appreciate your feedback.
> > These numbers refer to the local convergence rate:
> >  $ 1 $ refers to a linear rate,
> >  $ 1+ $ refers to a super-linear rate,
> >  $ 2 $ refers to a quadratic rate.
> >
> > We have added this explanation to the caption of Table 1.
> >
> >
> >
> >
> > **(Q3) About the $ 1/s $ variance chosen for the elements of $ P $.**
> >
> > **Response:**
> > We would like to point out that the choice of variance in the normal distribution is not critical for the algorithm, and we could have chosen a unit variance instead of $ 1/s $. Changing the variance in $ P $ only affects the value of the step size $ \eta_k $, which needs to be scaled accordingly. We specifically chose $ 1/s $ so that the Johnson-Lindenstrauss Lemma (Lemma B.1) can be stated without any scaling on the matrix $ P $.
> >
> >
> >
> >
> > **(Q4) Explain the "homogenization trick".**
> >
> > **Response:**
> > We have added an explanation about the homogenization trick in Appendix A, just below problem (A.0.1).
> >
> >
> >
> >
> > **(Q5) Explain which eigensolver is used in the experimental section for the subproblem.**
> >
> > **Response:**
> > Thank you for pointing this out. To satisfy the conditions imposed by HSODM, we used the Lanczos method in the numerical experiments. We have added this explanation at the beginning of Section 4.
> >
> >
> >
> >
> > **(Q6) Precision about the spatial complexity used in HSODM and RSHTR**
> >
> > **Response:**
> > It is true that the Lanczos method does not require explicitly forming the Hessian, and only Hessian-vector products (HVPs) are needed. However, since it is difficult to rigorously evaluate the spatial complexity of HVPs, we assume a general solver here and discuss both full-space and subspace algorithms assuming the Hessian is stored.
> >
> > By writing that the spatial complexity of the proposed method is $ O(sn) $, it might have caused a misunderstanding that the proposed method does not store the Hessian. However, the proposed method requires a spatial complexity of $ O(s^2) $ for storing the (restricted) Hessian. As a result, the spatial complexity for the random matrix, $ O(sn) $, becomes dominant. We have added a footnote (page 5) in the paper to clarify this.
> >
> > **(Q7) Clarify the relationship between the gradient norm and the relative function values.**
> >
> > **Response:**
> > We added further explanations in line 314 of the paper:
> > "Since we want to obtain an $ \varepsilon $-FOSP and not an $ \sqrt{n/s}\varepsilon $-FOSP, we need to scale down $ \varepsilon $ by $ \sqrt{s/n} $, leading to an increase in the number of iterations by a factor $ (n/s)^{3/4} $. We have added detailed explanations in Appendix D.3, just after the proof of Theorem 3.1."
> >
> > **(Q8) Explain the difference in the stopping criterion compared to HSODM in Zhang et al. (2022). Are they equivalent?**
> >
> > **Response:**
> > The stopping criteria are essentially the same, and we believe that the theoretical guarantees hold for RSHTR using the latest version of the stopping criterion as well. To confirm, we have also verified with the authors that the stopping criteria are fundamentally equivalent. We have added a footnote at the beginning of Appendix C to explain that; thank you for bringing this to our attention.
> >
> > **(Q9) Explain the notation $ \sqrt{\bar H} $.**
> >
> > **Response:**
> > We have added this to the list of notations in page 3.
> >
> > **Other points raised in the "Reference" paragraph**
> > **Response:**
> > Thank you for your thorough feedback.
> > We have addressed all of the points you raised.

---

> > ### Comment · Reviewer_zCBT · 2024-11-24
> >
> > Thank you for your detailed responses to every aspect, especially the supplementary explanations on the theory and related experiments. I believe the authors have made significant efforts. I would like to point out a minor issue: in the proof presented in Appendix C.2, the notation for the iteration, such as $j$ or $j_k$, should be consistent, and it should also be explicitly included in Algorithm 4.

---

> > > ### Author Response · Authors · 2024-11-24
> > >
> > > We have corrected it; it should be clearer to follow Section C.2 together with Algorithm 4.
> > > Again, thank you very much for helping us to improve the paper.

---

> > > > ### Comment · Reviewer_zCBT · 2024-11-25
> > > >
> > > > All my concerns have been resolved, and I will adjust my score accordingly.

---

> ### Comment · Reviewer_zCBT · 2024-11-24
>
> Thank you to the authors for their patient responses. All my questions have been addressed, and I appreciate the effort the authors have put in.

---

### Meta-Review · Area_Chair_BZsK · 2024-12-21

**Metareview:**

This paper presents a random subspace trust-region-based method for addressing nonconvex, unconstrained optimization problems. The paper is well-structured and makes a clear contribution to the nonconvex optimization literature. The authors demonstrate, for the first time, locally linear and quadratic convergence rates under certain assumptions. Empirical results showcase the practical advantages of the proposed algorithm compared to existing methods. The reviewers recognized the novelty and significance of this work and raised only minor concerns regarding readability and clarity of the theoretical results, all of which were addressed by the authors during the rebuttal phase. Therefore, I recommend  acceptance  of this paper.

**Additional Comments On Reviewer Discussion:**

During the rebuttal phase, the authors addressed reviewer zCBT’s concerns by fixing typos and providing clarifications on numerical constants and other technical aspects of their theoretical results. To respond to reviewer hJM7’s concerns regarding the empirical evaluation, they added additional experiments comparing their proposed algorithm with Adam and AdaGrad on the MNIST and CIFAR-10 datasets. Finally, they clarified certain technical details to address reviewer qDh3’s concerns and explained how the novelty of their proposed approach and theoretical results compare to related work. All of the authors’ responses were convincing, and the reviewers subsequently increased their scores.

---

### Decision · Program_Chairs · 2025-01-22

Accept (Spotlight)